

**Quantifying streamflow and active groundwater storage in response to climate**
**warming in an alpine catchment on the Tibetan Plateau**
Lu Lin[a,b], Man Gao[c], Jintao Liu[a,b*], Xi Chen[a,b,c], Hu Liu[d]
[a] *State Key Laboratory of Hydrology-Water Resources and Hydraulic Engineering,*
*Hohai University, Nanjing 210098, People's Republic of China*
[b] *College of Hydrology and Water Resources, Hohai University, Nanjing 210098,*
*People's Republic of China*
[c] *Institute of Surface-Earth System Science, Tianjin University, Tianjin 300072, People's*
*Republic of China*
[d] *Linze Inland River Basin Research Station, Chinese Ecosystem Research Network,*
*Lanzhou 730000, People's Republic of China*
*\* Corresponding author. Tel.: +86-025-83787803; Fax: +86-025-83786606.*
*E-mail address: jtliu@hhu.edu.cn (J.T. Liu).*





**Abstract**
Climate warming is changing streamflow regimes and groundwater storage in cold
alpine regions. In this study, a headwater catchment named Yangbajain in the Lhasa
River basin on the Tibetan Plateau is adopted as the study area for quantifying
streamflow changes and active groundwater storage in response to climate warming.
The catchment is characterized by alpine glacier and frozen ground which covers about
11% and 86% of the total area, respectively. The changes in streamflow regimes
(including quickflow and baseflow) and climate factors are evaluated based on hydro-
meteorological observations from 1979 to 2013. Then active groundwater storage in
autumn and early winter is quantified by recession flow analysis assuming
nonlinearized outflow from aquifers into streams. The results show that annual
streamflow increases significantly at a rate of about 12.30 mm/10a during this period.
The significant increase of annual air temperature compared with nonsignificant
variation of annual precipitation indicates that the climate warming takes
responsibilities to the increase of streamflow. It is believed that the increased
streamflow is mainly fed by glacier meltwater, which has led to over 25% loss of the
total glacial volume in the past 50 years (1960-2009) in this catchment. Moreover, the
significant increase of annual baseflow at a rate of about 10.95 mm/10a is the dominant
factor for the increase of the total streamflow. Through recession flow analysis, we find
that recession coefficient $K$ and active groundwater storage $S$ in autumn and early
winter increase significantly at the rates of about 7.70 $(mm^{0.79}d^{-0.21})$/10a and 19.32



mm/10a during these years. The increase of active groundwater storage can partly be
explained by frozen ground degradation, which lead to the enlargement of groundwater
storage capacity and accommodate more summer rainfall and meltwater in the wide
and flat valley, and then slowly release them into streams in the following seasons. Thus,
it is reasonable to attribute the increase of baseflow and the slowdown of baseflow
recession process in autumn and early winter to the enlargement of groundwater storage
capacity. Through quantifying streamflow changes and active groundwater storage in
response to warming-induced changes, this study provides a perspective to clarify the
way of glacial retreat and frozen ground degradation on hydrological processes.
**Keywords:** Climate warming; Streamflow; Groundwater storage; Glacier retreat;
Frozen ground degradation; Tibetan Plateau
**1. Introduction**

Often referred to as the "Water Tower of Asia", the Tibetan Plateau (TP) is the source

area of major rivers in Asia, e.g., the Yellow, Yangtze, Mekong, Salween, Indus, and
Brahmaputra Rivers (Cuo et al., 2014). The delayed release of water resources on the
TP through glacier melt can augment river runoff during dry periods as a pivotal role
for water supply for downstream populations, agriculture and industries in these rivers
(Viviroli et al., 2007; Pritchard, 2017). However, the TP is experiencing a significant
warming trend during the last half century (Kang et al., 2010; Liu and Chen, 2000).
Along with the rising temperature, major warming-induced changes have occurred over
the TP, such as glacier retreat (Yao et al., 2004; Yao et al., 2007) and frozen ground



degradation (Wu and Zhang, 2008). Hence, it is of great importance to elucidate how
climate warming influences hydrological processes and water resources on the TP.

In cold alpine catchments, glacier is known as "solid reservoir" that supplies water

as streamflow, while frozen ground, especially permafrost, servers as an impermeable
barrier to the interaction between surface water and groundwater (Immerzeel et al.,
2010; Walvoord and Kurylyk, 2016). Since the 1990s, most glaciers across the TP have
retreated rapidly due to global warming and caused an increase of more than 5.5% in
river runoff from the plateau (Yao et al., 2007). Meltwater is the key contributor to
streamflow increase especially for headwater catchments with larger glacier coverage
(>5%) (Bibi et al., 2018). Meanwhile, in a warming climate, numerous studies
suggested that frozen ground on the TP has experienced a noticeable degradation during
the past decades (Cheng and Wu, 2007; Wu and Zhang, 2008). Frozen ground
degradation can modify surface conditions and change thawed active layer storage
capacity in the alpine catchments (Niu et al., 2011). Thawing of frozen ground increases
surface water infiltration, supports deeper groundwater flow paths, and then enlarges
groundwater storage, which is expected to have a profound effect on flow regimes
(Bense et al., 2009; Bense et al., 2012; Walvoord and Striegl, 2007; Woo et al., 2008;
Ge et al., 2011; Walvoord and Kurylyk, 2016). In cold alpine catchments where large
areas of glacier and frozen ground exist, warming-induced glacier and frozen ground
co-variations fundamentally affect the water supply and the mechanisms of streamflow
generation and change (Cuo et al., 2014; Pritchard, 2017).



It is challenging to understand how glacier melt and frozen ground thaw alters the
mechanism of streamflow in a warmer climate due to the complicated interactions
between hydrological and cryospheric processes. In earlier phase of glacier melt,
accelerated glacier retreat will bring large quantities of meltwater available directly for
surface runoff or indirectly for groundwater recharge (Bayard et al., 2005). Meanwhile,
frozen ground thawing may allow for increased groundwater recharge from meltwater
infiltration (Evans and Ge, 2017). Generally, climate warming is hypothesized to
generate a quantitative and temporal shift in the partitioning of meltwater between
surface runoff and groundwater flow, and thereby alter the quantity and timing of
baseflow (Green et al., 2011; Evans et al., 2018). Evans et al. (2015) found that an
increase in mean annual surface temperature of 2°C reduced approximately 28% areal
extent of permafrost and tripled baseflow contribution to streamflow using a physically
based groundwater model in a headwater catchment of the Heihe River on the northern
TP. Qin et al. (2016) discovered that the increasing precipitation and the thawing of
frozen ground were the main factors on the increase of baseflow with no significant
change in surface runoff in the upper Heihe River basin of the northeastern TP. Previous
data-based studies indicated that the baseflow has increased especially during winter
with a reduction or no pervasive change in summer streamflow in the central and
northern TP (Liu et al., 2011; Niu et al., 2016) as well as the Arctic rivers (Walvoord
and Striegl, 2007; Smith et al., 2007; St. Jacques and Sauchyn, 2009). Moreover, Bense
et al. (2012) suggested that the increasing groundwater storage caused by frozen ground



degradation would delay baseflow increase possibly by several decades to centuries
based on numerical simulations. The slowdown in baseflow recession processes was
found in the northeastern and central TP (Niu et al., 2011; Niu et al., 2016; Wang et al.,
2017), in northeastern China (Duan et al., 2017), and in Arctic rivers (Lyon et al., 2009;
Lyon and Destouni, 2010; Walvoord and Kurylyk, 2016).
While, previous qualitatively studies were important for understanding the effects of
climate warming on hydrological changes in cold alpine catchments (Niu et al., 2011;
Niu et al., 2016; Wang et al., 2017). However, quantitatively characterizing storage
properties and sensitivity to climate warming in cold alpine catchments is important for
local water as well as downstream water management (Staudinger, 2017). Moreover,
revealing the storage characteristics makes it easier to predict hydrological cycle and
streamflow changes response to warming climate in cold alpine catchments (Singleton
and Moran, 2010). Thus, this study focuses on quantifying streamflow and aquifer
storage volume response to changes in glacier melt and frozen ground thaw at
catchment scale on the southern TP. Given the difficulty of direct measurements for
catchment aquifer storage (Staudinger, 2017; Käser and Hunkeler, 2016) and low
spatial resolution for the GRACE satellites to assess total groundwater storage changes
at catchment scale (Green et al., 2011), an alternative method, namely, recession flow
analysis, can be theoretically used to derive the active groundwater storage volume in
the phreatic aquifer to reflect frozen ground degradation in a catchment (Brutsaert and
Nieber, 1977; Brutsaert et al., 2008). For example, the groundwater storage changes



have been inferred by recession flow analysis assuming linearized outflow from
aquifers into streams (Lin and Yeh, 2017). However, the non-linear of the storage
discharge relationship dominates baseflow recession processes for most catchments due
to the complex structures and properties of catchment aquifers (Chapman, 1999; Liu et
al., 2016). Moreover, groundwater storage computed by assuming the aquifers as linear
reservoir cannot reflect the actual storage (Wittenberg, 1999). Lyon et al. (2009)
adopted the non-linear reservoir to fit flow recession curves for derivation of aquifer
attributes, which can be developed for inferring aquifer storage. Buttle (2017) used
Kirchner (2009) approach for estimating dynamic storage in different basins and found
that storage and release of dynamic storage may mediate baseflow response to temporal
changes.
In this study, the long-term changes in streamflow and climate factors in a glacier-
fed headwater catchment with frozen ground in the Lhasa River basin of the south-
central TP is analyzed using non-parametric tests during the period 1979-2013. The
First and Second Glacier Inventory of China is used to assess the response of glacier
variations to climate warming. Changes in streamflow components, baseflow recession
process and active groundwater storage are examined. The main objectives of this study
are (1) to identify the water source for streamflow changes in climate warming; (2) to
discuss the water volume changes in the partitioning between surface runoff and
groundwater flow due to changes in glacier melt and frozen ground thaw; (3) to quantify
active groundwater storage volume by recession flow analysis assuming nonlinearized



outflow from aquifers into streams, and to analyze the impacts of the changes in active
groundwater storage on streamflow variation.
**2. Materials and Methods**
**2.1. Study area**
Located on the south-central TP, the Yangbajain catchment is a glacier-fed headwater
catchment with highly frozen ground coverage in the western part of the Lhasa River
Basin (Figure 1a). The catchment has an area of approximately 2,645 km$^2$ and its
elevations range from 4,270 to 6,400 m (Figure 1b). In the east of the catchment, the
wide and flat valley (Figure 1b) is located in the Damxung-Yangbajain fault of the
southeastern piedmont of Nyainqêntanglha Mountains (Jiang et al., 2016; Yang et al.,
2017) with low-lying flat terrain and thicker aquifers due to the great thickness
quaternary loose sediment (Wu and Zhao, 2006). The coverage of glacier area is about
11% in the catchment, which is the highest glacierized sub-catchment in the Lhasa
River Basin. The total glacier area was about 316.31 km$^2$ in 1960 according to the First
Chinese Glacier Inventory (Mi et al., 2002) and most glaciers were found along the
Nyainqêntanglha Mountains range (Figure 1c). The ablation period of the glaciers
ranges from June to September with the glacier termini at about 5,200 m (Liu et al.,
2011). According to the new map of permafrost distribution on the TP (Zou et al., 2017),
the wide and flat valley is underlain by seasonally frozen ground (Figure 1c). It is
estimated that seasonally frozen ground and permafrost accounts for about 64% and 22%
of the total catchment area, respectively. The lower limit of alpine permafrost is around



4,800 m, and the thickness of permafrost varies from 5 m to 100 m (Zhou et al., 2000).
The climate in the catchment is characterized by semi-arid temperate monsoon
climate. The average annual air temperature of the Yangbajain catchment is
approximately -2.3°C with monthly variation from -8.6°C in January to 3.1°C in July
(Figure 2). The average annual precipitation at the Yangbajain station (4,305 m) in the
valley is about 427 mm. The intra-annual distribution of precipitation is extremely
uneven due to the pronounced rainy season during the summer monsoon (June-August)
and the dry season lasting the rest of the year. Nearly 73% of the total precipitation
occurs in summer, while only 1% of the precipitation occurs in winter (December-
February) (Figure 2).
The average annual runoff depth is 277.7 mm, and the intra-annual distribution of
streamflow is uneven. Approximately 63% of the annual streamflow is observed in
summer, whereas in the winter season, streamflow is low and accounts for only 4% of
the annual streamflow (Figure 2). Streamflow is recharged mainly by monsoon rainfall
and summer meltwater. The river in winter is only recharged by groundwater, which is
greatly affected by the freeze-thaw cycle of frozen ground and the active layer (Liu et
al., 2011).
**2.2. Data**
Daily streamflow and precipitation data at the Yangbajain station (4,305 m) during
the period 1979-2013 are collected from Tibet Autonomous Region Hydrology and
Water Resources Survey Bureau. The monthly meteorological data at the Damxung



station (4,289 m), which is neighbor to the Yangbajain catchment (Figure 1a), are
obtained    from    the    China    Meteorological    Data    Sharing    Service    System
(http://data.cma.cn/) for the years from 1979 to 2013. In this study, the method of
meteorological data extrapolation by Prasch et al. (2013) is adopted to obtain the
discretisized air temperature (with cell size as 1 km×1 km) of the Yangbajain catchment
based on the air temperature of the Damxung station assuming a linear lapse rate. The
mean monthly lapse rate is set to 0.44 °C 100 m$^{-1}$ with elevation below 4,965 m and
0.78 °C 100 m$^{-1}$ with elevation above 4,965 m in the catchment (Wang et al., 2015).

The glacier and frozen ground data are provided by the Cold and Arid Regions

Science Data Center at Lanzhou (http://westdc.westgis.ac.cn/). The distribution, area
and volume of glacier are based on the First and Second Chinese Glacier Inventory in
1960 and 2009. The distribution and classification of frozen ground are collected from
a new map of permafrost distribution on the Tibetan Plateau (Zou et al., 2017).
**2.3. Methods**
*2.3.1. Mann-Kendall test with trend free pre-whitening*

The Mann-Kendall (MK) test is applied to detect trends of hydro-meteorological time

series, which is robust against outliers and is suitable for data with non-normally
distributed or non-linear trends (Mann, 1945; Kendall, 1975). To remove the serial
correlation from the examined time series, a Trend-Free Pre-Whitening (TFPW)
procedure is needed prior to applying the MK test (Yue et al., 2002). A more detailed
description of the Trend-Free Pre-Whitening (TFPW) approach was provided by Yue



et al. (2002).
The MK test statistic $s$ is calculated as
$$s = \sum_{i=1}^{n-1} \sum_{j=i+1}^{n} \mathrm{sgn}\left(x_j - x_i\right)$$
(1)

where, $x_j$ and $x_i$ are the data values in sequence, $n$ is the sequence length, and sgn ($x_j$-
$x_i$) are recorded as
$$\mathrm{sgn}\left(x_j - x_i\right) = \begin{cases} 1, & x_j > x_i \\ 0, & x_j = x_i \\ -1, & x_j < x_i \end{cases}$$
(2)

The variance of $s$ is proposed by the equation (3)
$$\mathrm{Var}(s) = \frac{n(n-1)(2n+5)}{18}$$
(3)

Then, the standardized test statistic $Z_C$ can be transformed from statistical value $s$,
and is computed by equation (4)
$$Z_C = \begin{cases} \dfrac{s-1}{\sqrt{\mathrm{Var}(s)}} & s>0 \\ 0 & s=0 \\ \dfrac{s+1}{\sqrt{\mathrm{Var}(s)}} & s<0 \end{cases}$$
(4)

When $|Z_C| \leq 1.96$, there is no significant trend. The trend is at the 5% significance
level if $|Z_C| > 1.96$, and at the 1% significance level if $|Z_C| > 2.58$. A positive value of
$Z_C$ indicates an upward trend, whereas a negative value indicates a downward trend in
the tested time series.
The trend magnitude is computed by Theil-Sen estimator (Sen, 1968)
$$\beta = \mathrm{median}\left(\frac{x_i - x_j}{i - j}\right), \forall j < i$$
(5)



where $1<j<i<n$, a positive value of $\beta$ indicates an upward trend, and a negative value
indicates a downward trend.
*2.3.2. Baseflow separation*

In this paper, the most widely used one-parameter digital filtering algorithms is

adopted for baseflow separation (Lyne and Hollick, 1979). The first filter equation is
expressed as
$$q_t = \alpha q_{t-1} + \frac{1+\alpha}{2}\left(Q_t - Q_{t-1}\right) \qquad (6)$$

$$b_t = Q_t - q_t \qquad (7)$$

where $q_t$ and $q_{t-1}$ are the filtered quickflow at time step $t$ and $t$-1, respectively; $Q_t$ and
$Q_{t-1}$ are the total runoff at time step $t$ and $t$-1; $b_t$ is the filtered baseflow. $\alpha$ is the filter
parameter, ranging from 0.9 to 0.95.
*2.3.3. Determination of active groundwater storage*

The method of recession flow analysis is widely used to investigate the baseflow

recession characteristics and the storage discharge relationship of catchments (Gao et
al., 2017). Physical considerations based on hydraulic groundwater theory suggest that
the groundwater storage in a catchment can be approximated as a power function of
baseflow rate at the catchment outlet (Brutsaert, 2008)
$$S = Ky^m \qquad (8)$$

where $y$ is the rate of baseflow in the stream in a catchment, $S$ is the volume of active
groundwater storage in the catchment aquifers (see in Figure 3), abbreviated as
groundwater storage in the following context. And $K$ and $m$ are constants depending on



the catchment physical characteristics. $K$ represents the time scale of the catchment
streamflow recession process, commonly referred to as baseflow recession coefficient.
During a period without precipitation and evapotranspiration, the flow in a stream
can be assumed to depend solely on the groundwater storage from the upstream aquifers.
For such baseflow conditions, the conservation of mass equation can be represented as
$$\frac{dS}{dt} = -y \tag{9}$$
where $t$ is the time. Substitution of equation (8) in equation (9) yields
$$-\frac{dy}{dt} = ay^b \tag{10}$$
where $dy/dt$ is the temporal change of the baseflow rate during recessions, and the
constants $a$ and $b$ are called the recession intercept and recession slope of plots of $-dy/dt$
versus $y$ in log-log space, respectively. The parameters of $K$ and $m$ in equation (8) can
be expressed by $a$ and $b$, where $K = 1/\left[a(2-b)\right]$ and $m = 2-b$. In the storage
discharge relationship, the aquifer responds as a linear reservoir if $b$=1, and as non-
linear reservoir if $b$≠1.
In our study, the baseflow recession data are selected from the streamflow
hydrographs, which remarkably decline for at least 3 days after rainfall ceases and
remove the first 2 days to avoid the impact of storm flow (Brutsaert and Lopez, 1998).
A variable time interval $\Delta t$ is used to properly scale the observed drop in streamflow to
avoid discretization errors on $-dy/dt$~$y$ plot due to measurement noise, especially in the
log-log space (Rupp and Selker, 2006; Kirchner, 2009). Meanwhile, the difference of
baseflow $\Delta y$ in the catchment exceeds a critical precision threshold $\Delta y_{crit}$ of 0.02





mm/day. Then the constants *a* and *b* are fit by using a non-linear least squares through
all data points of $-dy/dt$ versus *y* in log-log space for all years (1979-2013) to avoid the
difficulty of defining a lower envelop of the scattered points (Lyon et al., 2009). With
the fixed slope *b* during recessions (i.e., $b \neq 1$ remains constant), it should be possible to
observe changes in catchment aquifer properties by fitting the intercept *a* as a variable
across different years. Since the values of *K* and *m* for each year can be calculated by
fitting recession intercept *a* and the fixed slope *b*, the groundwater storage *S* in a
catchment is obtained through equation (8) based on average rate of baseflow during
recessions.
**3. Results and Discussion**
**3.1. Variation of annual streamflow and its components**
The annual streamflow of the Yangbajain catchment shows an increasing trend at the
5% significance level with a mean rate of about 12.30 mm/10a over the period 1979-
2013 (Table 1 and Figure 4a). Meanwhile, annual mean air temperature exhibits an
increasing trend at the 1% significance level with a mean rate of about 0.28 °C/10a
(Table 1 and Figure 5a). However, annual precipitation has nonsignificant trend during
this period (Table 1 and Figure 5b). The similar variation trends between annual
streamflow and annual air temperature indicate that the changes of air temperature may
act as a primary climatic factor for streamflow increase.
As the significant rising of air temperature, glacier in the catchment has been
retreating continuously. According to the twice Chinese Glacier Inventory (I & II





volume) in 1960 and 2009, the total glacial area and volume have decreased by 38.06
km$^2$ (12.0%) and 0.47×10$^{10}$ m$^3$ (26.2%) over the past 50 years (Figure 6). With the
nonsignificant increase of annual precipitation, it is reasonable to attribute annual
streamflow increase to the accelerated glacier retreat as the consequence of increasing
annual air temperature. This conclusion is also consistent with previous results by
Prasch et al. (2013), who suggested that glacial meltwater contribution to streamflow
would remain increase in the Yangbajain catchment together with significant increase
in streamflow and nonsignificant trend in precipitation by quantifying present and
future glacier meltwater contribution to runoff.
Overall, the annual mean baseflow contributes about 59% of annual mean
streamflow in the catchment through baseflow separation method. As annual
streamflow increases significantly, it is necessary to analyze to what extent the changes
in two streamflow components lead to streamflow increase. The result shows that
annual baseflow exhibits a significant increasing trend at the 1% level with a mean rate
of about 10.95 mm/10a over the period 1979-2013 (Table 1 and Figure 4b). This trend
is statistically nonsignificant for annual quickflow during the period (Table 1). Thus,
the increase in baseflow is the main contributor to streamflow increase. It can be further
concluded that streamflow is recharged by the increased meltwater from the accelerated
glacier retreat which may be partly stored in soil and aquifers in the wide and flat valley
(Figure 1b), and subsequently discharge into streams as baseflow.





### 3.2. Variation of seasonal streamflow and its components


The hydrograph of the Yangbajing catchment shows obvious intra-annual variation
(Figure 2). Streamflow sources and main components also change with the streamflow
magnitude. The variation trends of streamflow regimes also change across seasons. In
autumn, winter, and spring, both streamflow and baseflow show significant increasing
trends at least at the 5% level (Figures 7c, 7d and 7a). However, quickflow exhibits
nonsignificant trend for all seasons (Table 1). Streamflow increases significantly at the
5% level in autumn and the increasing trends reach the significant level of 1% in winter
and spring. Baseflow increases significantly at the 1% level in spring and autumn and
the increasing trend is at the 5% significance level in winter. However, the trends are
not statistically significant for both streamflow and its two components (quickflow and
baseflow) in summer (Figure 7b). As to the meteorological factors, mean air
temperature in all seasons increase significantly at the 1% level especially during winter
with the rate of about 0.51°C/10a (Table 1 and Figure 8), whereas precipitation in each
season shows nonsignificant trend during these years (Table 1).
Compared with monsoon rainfall as the main water source for summer which
accounts for about 73% of the total precipitation in the whole year, the corresponding
meltwater from glacier is considerable but its contribution to streamflow is limited.
Moreover, the summer meltwater and rainfall will partly infiltrate into soils and aquifers.
Carey and Quinton (2004) suggests that in snow and permafrost catchments with the
thin river valley and the steep slopes, meltwater infiltrates soils and resides in temporary



storage at the beginning of the melt period, and then are allowed to rapidly drain through
surface layers. However, due to thicker aquifers in the wide and flat catchment valley
(Figure 1b), summer meltwater and rainfall stored in aquifers are allowed to release
slowly from groundwater storage as baseflow in the following seasons, which has led
to the stability of baseflow in summer and the significant increase of baseflow in
autumn, winter and spring.
**3.3. Variation of baseflow recession rate and groundwater storage**
Using the data selected procedure mentioned in the section 2.3.3, we adopted daily
streamflow and precipitation records from September to December (the autumn and
early winter) over the period 1979-2013 in the catchment, during which the hydrograph
with little precipitation usually declines consecutively and smoothly. The fitted slope $b$
is equal to 1.79 through the non-linear least square fit of equation (10) for all data points
of $-dy/dt$ versus $y$ in log-log space during the period 1979-2013. With the fixed slope
$b=1.79$, the recession coefficient $K$ and groundwater storage $S$ can be quantified by all
decades of the 1980s, 1990s and 2000s, and year-to-year from 1979 to 2013. For each
decade, the recession intercept $a$ could be fitted by the fixed slope $b=1.79$. Then, the
values of $K$ and $m$ for each decade can be determined with the fitted recession intercept
$a$ and the fixed slope $b$. And the groundwater storage $S$ for each decade can be directly
estimated from the average rate of baseflow during recession period and the values of
$K$ and $m$ through equation (8). Meanwhile, the recession coefficient $K$ and groundwater
storage $S$ for each year can also be calculated according to the above procedure.





Figure 9 shows the non-linear least square fit of equation (10) to the plot of $-dy/dt$
versus $y$ in log-log space for all recession data points of the observation records for each
decade of the 1980s, 1990s and 2000s, respectively. As shown in Figure 9, the recession
data points and fitted recession curves of each decade gradually move downward as
time goes on. It indicates that, with the fixed slope $b$, the recession intercept $a$ gradually
decreases and recession coefficient $K$ gradually increases. The values of recession
coefficient $K$ for each decade are respectively 77 mm$^{0.79}$d$^{-0.21}$, 84 mm$^{0.79}$d$^{-0.21}$ and 103
mm$^{0.79}$d$^{-0.21}$ in the 1980s, 1990s and 2000s through recession flow analysis, which is
consistent with the results in Figure 9. Figure 10a shows the inter-annual variation of
recession coefficient $K$ during the period 1979-2013. The recession coefficient $K$
increases slowly in the 1980s, fluctuates slightly in the 1990s and increases rapidly in
the 2000s. But its overall increasing trend is similar to the results obtained from decades
analysis. The trend of recession coefficient $K$ shows significant increase at the 5% level
at a rate of about 7.70 (mm$^{0.79}$d$^{-0.21}$)/10a from 1979 to 2013. This long-term variation
of recession coefficient $K$ from September to December indicates that baseflow
recession process during autumn and early winter gradually slows down in the
catchment.
The mean values of groundwater storage $S$ for each decade are 130 mm, 148 mm and
188 mm in the 1980s, 1990s and 2000s, respectively. The trend analysis suggests that
the groundwater storage $S$ shows an increasing trend at the 5% significance level with
a rate of about 19.32 mm/10a during the period 1979-2013 (Figure 10b). It indicates



that groundwater storage has been enlarged during autumn and early winter. The long-
term trend of groundwater storage $S$ from 1979 to 2013 is consistent with the values
across decades. The inter-annual variation of groundwater storage $S$ is also similar with
recession coefficient $K$ (Figure 10a and 10b).

The increased groundwater storage $S$ in autumn and early winter is associated with

the hypothesis that frozen ground degradation due to the significant rising air
temperature during autumn and winter (Figure 8c and 8d), which can enlarge
groundwater storage capacity (Niu et al., 2016). Figure 3 depicts the changes of surface
flow and groundwater flow paths in a glacier-fed and underlying-frozen ground
catchment under past climate and warmer climate, respectively. As frozen ground
extent continues to decline and active layer thickness continues to increase in the wide
and flat valley, the enlargement of groundwater storage capacity can provide enough
storage space to accommodate increasing meltwater, and support more meltwater to
percolate into deeper aquifers rather than surface layers, and thereby increase
groundwater storage in the valley floor (Figure 3). Then, the increase of groundwater
storage in autumn and earlier winter allows more groundwater discharge into streams
as baseflow, and lengthens the time scale of the baseflow recession process indicated
by recession coefficient $K$. This leads to increase baseflow and slow baseflow recession
processes in autumn and early winter, as is shown in Figure 7c, 7d and Figure 10a. In
the late winter and spring, the increase of baseflow (Figure 7d and 7a) can be explained
by the delayed release of increased groundwater storage.



### 4. Conclusions


In this study, the changes of hydro-meteorological variables were evaluated to
identify the main climatic factor for streamflow increase during the period 1979-2013
of the Yangbajain catchment, a sub-catchment with larger glacierization and large-scale
frozen ground in the Lhasa River basin in the south-central TP. We analyzed the changes
of streamflow components through baseflow separation method. We quantified
baseflow recession process and active groundwater storage in autumn and early winter
by recession flow analysis assuming nonlinearized outflow from aquifers into streams,
and analyzed the seasonal variations of streamflow and its components in response to
the changes in active groundwater storage.
We find that the increase of annual streamflow is mainly due to the increase of annual
baseflow, which is caused by increased temperature rather than precipitation in the
long-term period. The decreased glacial volume due to climate warming has supplied
large quantities of glacial meltwater which recharges aquifers and resides in temporary
storage during summer, and then releases as baseflow during the following seasons.
Moreover, the increase of active groundwater storage in autumn and early winter can
partly be attributed to the enlargement of groundwater storage capacity by frozen
ground degradation, which can provide storage spaces for increased glacial meltwater.
This can partly explain why baseflow volume increases and baseflow recession process
slows down in autumn, winter, and spring seasons.
This study provides a fundamental understanding of the changes in streamflow and





groundwater storage under warming climate. It is of great importance to predict the
effects of future climate changes on water resources and hydrological processes in
highly glacier-fed and large-scale frozen ground regions. Further analysis is needed to
quantify summer meltwater contribution to streamflow, and to explore the change of
groundwater storage capacity as frozen ground continues to degrade.
**Acknowledgements:**
This work was supported by the National Natural Science Foundation of China
(NSFC) (grants 91647108, 91747203), the Science and Technology Program of Tibet
Autonomous Region (2015XZ01432), and the Special Fund of the State Key
Laboratory of Hydrology-Water Resources and Hydraulic Engineering (no

20185044312).

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






**Table 1.** Mann-Kendall trend test with trend-free pre-whitening of seasonal and annual mean air temperature (°C), precipitation (mm), streamflow (mm), baseflow (mm) and quickflow (mm) from 1979 to 2013.

| | Air temperature | | Precipitation | | Streamflow | | Baseflow | | Quickflow | |
|---|---|---|---|---|---|---|---|---|---|---|
| | $Z_C$ | $\beta$ (°C/a) | $Z_C$ | $\beta$ (mm/a) | $Z_C$ | $\beta$ (mm/a) | $Z_C$ | $\beta$ (mm/a) | $Z_C$ | $\beta$ (mm/a) |
| Spring | 2.73** | 0.026 | 0.90 | 0.290 | 3.05** | 0.206 | 2.99** | 0.147 | 0.98 | 0.042 |
| Summer | 2.63** | 0.013 | 1.30 | 2.139 | 0.92 | 0.549 | 1.27 | 0.429 | 0.50 | 0.128 |
| Autumn | 2.65** | 0.024 | -0.68 | -0.395 | 2.46* | 0.546 | 2.96** | 0.476 | 0.80 | 0.074 |
| Winter | 3.49** | 0.051 | -0.46 | -0.014 | 3.08** | 0.204 | 2.13* | 0.145 | 1.39 | 0.016 |
| Annual | 4.48** | 0.028 | 1.28 | 2.541 | 2.07* | 1.230 | 2.70** | 1.095 | 0.77 | 0.327 |

Comment: the symbols of asterisks *and ** mean statistically significant at the levels of 5% and 1%, respectively.






**Figure captions**
**Figure 1.** (a) The location, (b) elevation distribution, and (c) glacier and frozen ground
distribution (Zou et al., 2017) in the Yangbajain catchment of the Lhasa River basin in
the TP.
**Figure 2.** Seasonal variation of runoff depth ($R$), mean air temperature ($T$), and
precipitation ($P$) in the Yangbajain catchment.
**Figure 3.** Diagram depicting surface flow and groundwater flow due to glacier melt
and frozen ground thaw under (a) past climate and (b) warmer climate. Blue lines with
arrows are conceptual surface flow paths. Dark blue lines with arrows are conceptual
groundwater flow paths (after Evans and Ge. (2017)).
**Figure 4.** Variations of annual (a) runoff and (b) baseflow depth from 1979 to 2013.
**Figure 5.** Variations of annual (a) mean air temperature and (b) precipitation from 1979
to 2013.
**Figure 6.** The total area and volume of glaciers in the Yangbajain catchment in 1960
and 2009.
**Figure 7.** Variations of seasonal runoff and baseflow depth in (a) spring, (b) summer,
(c) autumn, and (d) winter from 1979 to 2013.
**Figure 8.** Variations of seasonal mean air temperature in (a) spring, (b) summer, (c)
autumn, and (d) winter from 1979 to 2013.
**Figure 9.** Recession data points of $-dy/dt$ versus $y$ and fitted recession curves by decades
in log-log space. The black point line, dotted line, and solid line represent recession



curves in the 1980s, 1990s, and 2000s, respectively.
**Figure 10.** Variations of (a) the recession coefficient $K$ and (b) groundwater storage $S$
from 1979 to 2013.





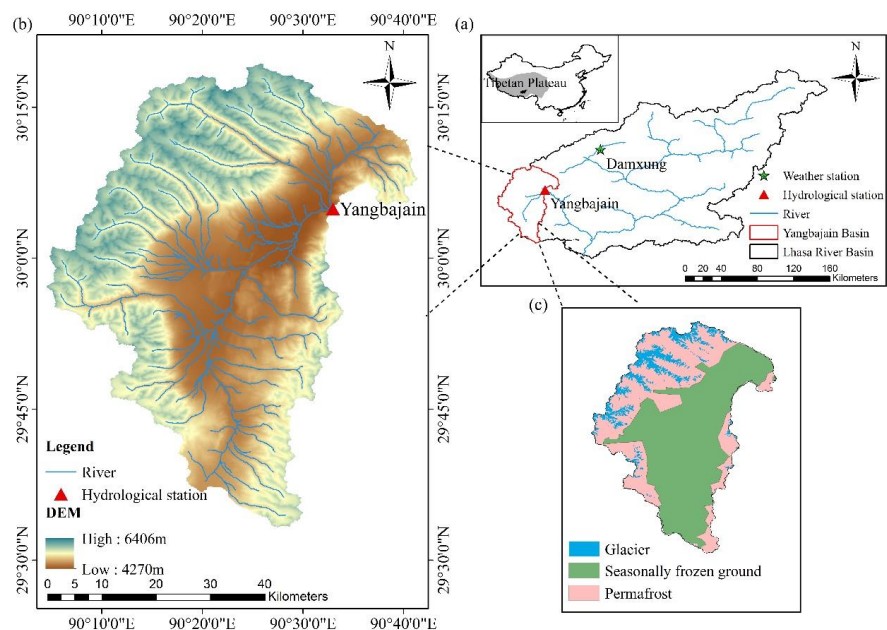

**Figure 1.** (a) The location, (b) elevation distribution, and (c) glacier and frozen ground

distribution (Zou et al., 2017) in the Yangbajain catchment of the Lhasa River basin in

the TP.



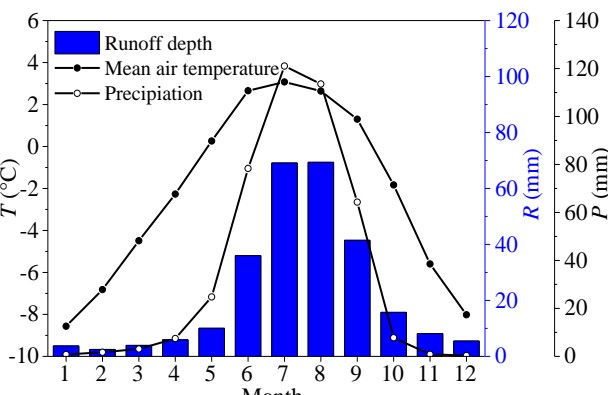

**Figure 2.** Seasonal variation of runoff depth ($R$), mean air temperature ($T$), and

precipitation ($P$) in the Yangbajain catchment.

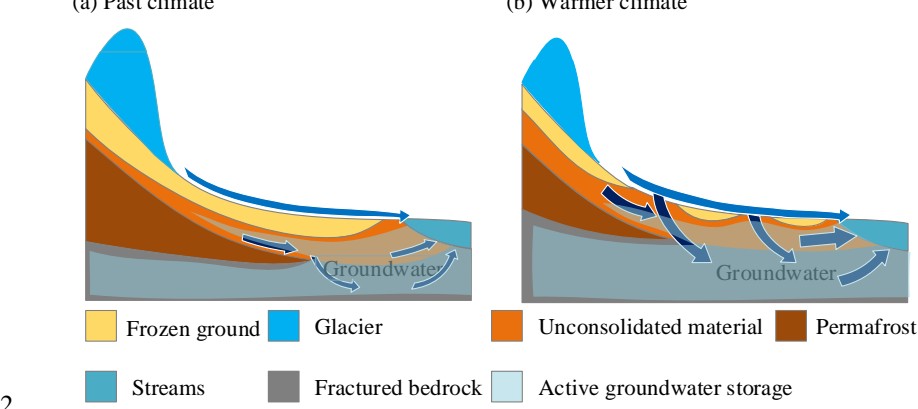

**Figure 3.** Diagram depicting surface flow and groundwater flow due to glacier melt

and frozen ground thaw under (a) past climate and (b) warmer climate. Blue lines with

arrows are conceptual surface flow paths. Dark blue lines with arrows are conceptual

groundwater flow paths (after Evans and Ge. (2017)).






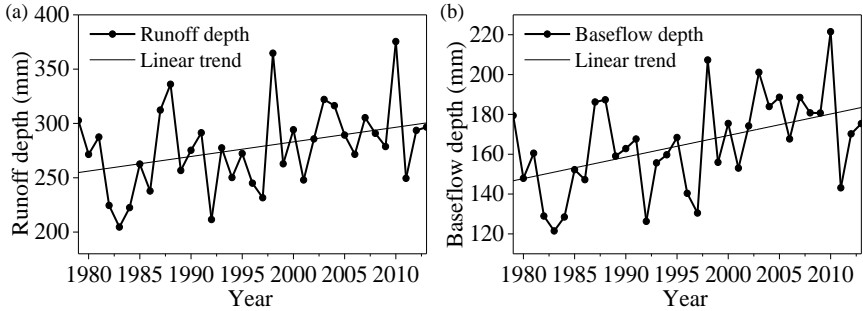


**Figure 4.** Variations of annual (a) runoff and (b) baseflow depth from 1979 to 2013.


**Figure 5.** Variations of annual (a) mean air temperature and (b) precipitation from

1979 to 2013.





**Figure 6.** The total area and volume of glaciers in the Yangbajain catchment in
1960 and 2009.

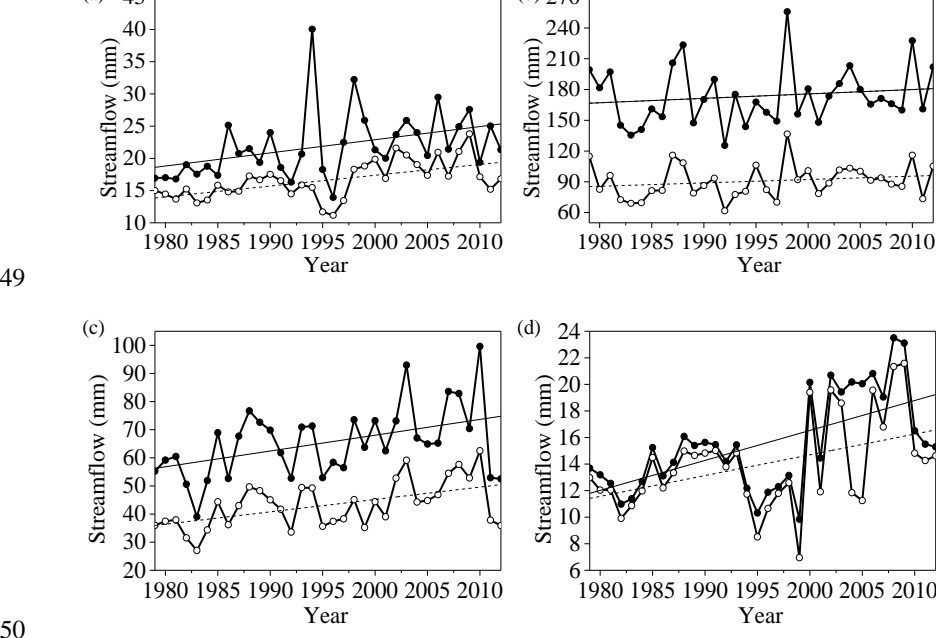



**Figure 7.** Variations of seasonal runoff and baseflow depth in (a) spring, (b)
summer, (c) autumn, and (d) winter from 1979 to 2013.





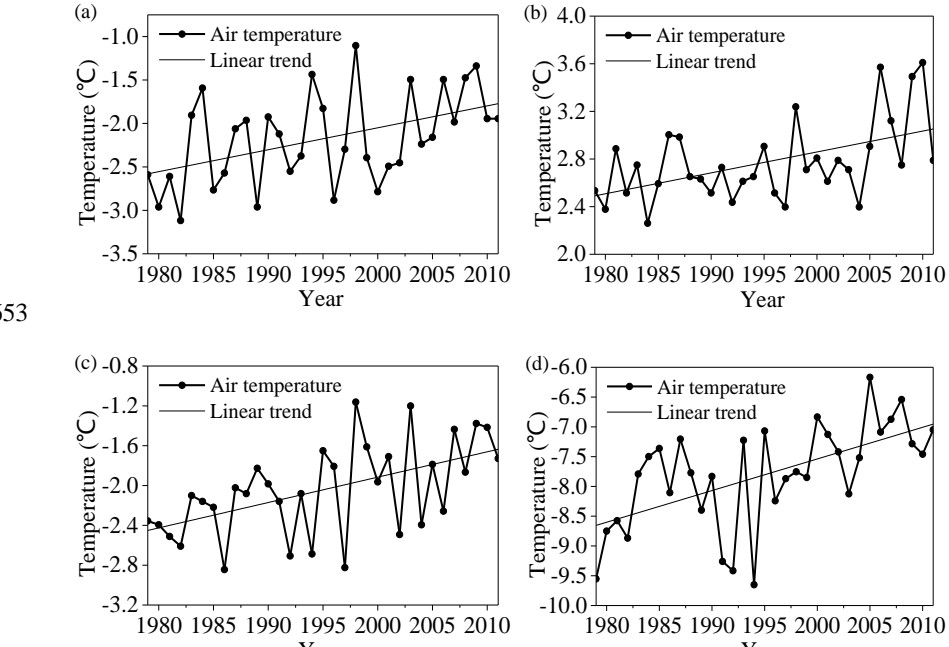



**Figure 8.** Variations of seasonal mean air temperature in (a) spring, (b) summer, (c)

autumn, and (d) winter from 1979 to 2013.




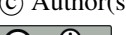



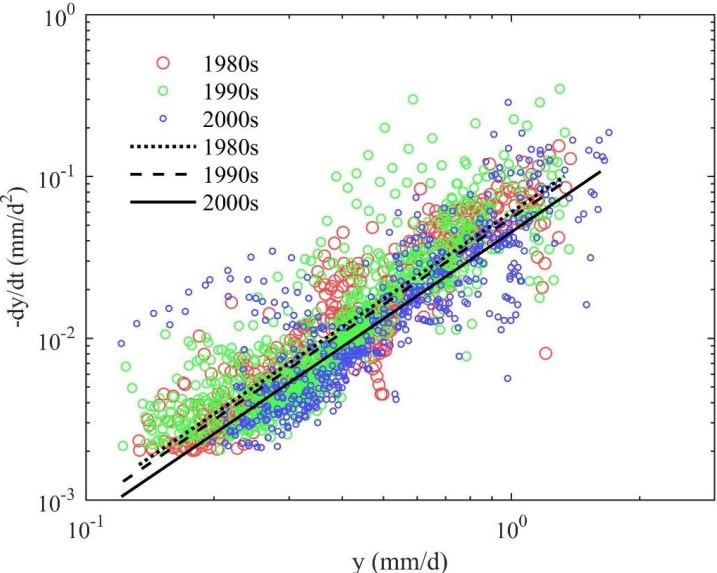

**Figure 9.** Recession data points of -$dy/dt$ versus $y$ and fitted recession curves by decades

in log-log space. The black point line, dotted line, and solid line represent recession

curves in the 1980s, 1990s, and 2000s, respectively.

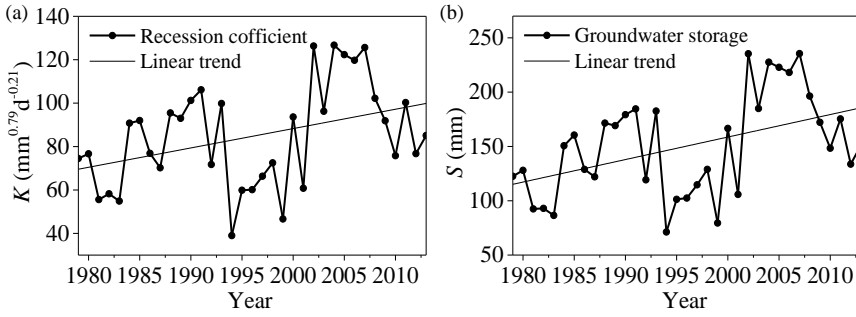

**Figure 10.** Variations of (a) the recession coefficient $K$ and (b) groundwater storage

$S$ from 1979 to 2013.