# Peer review of "Quantifying streamflow and active groundwater storage in response to climate"

_Hydrology and Earth System Sciences, 2018_

## Short Comment (SC1) · 10 Dec 2018

[supplement omitted: unrelated document]

---

## Referee Comment (RC1) · Anonymous Referee #1 · 3 Jan 2019

Review comments for the HESS Manuscript "Quantifying streamflow and active groundwater storage in response to climate warming in an alpine catchment on the Tibetan Plateau" by Line et al.

General comments: This work intended to quantify streamflow and aquifer storage volume response to changes in glacier melt and frozen ground thaw at a branch of Lhasa River Basin on the southern TP. However, the work is mostly in a qualitative way and lacks of in-depth quantitative analysis. Therefore, the conclusions are not supported by solid analysis or materials. For instance, in Abstract, lines 29-30, the statement " It is believed that the increased streamflow is mainly fed by glacier meltwater, ...". Firstly,

there is no quantitative analysis to support this statement in the manuscript. Secondly, in research paper, every statement or conclusion must be supported by solid analysis. How can authors use words like "It is believed..."?

The same issues exist everywhere in the text, other examples are, the sentences in Lines 36-42 are mostly qualitative descriptions or deductions instead of solid conclusions. Sentence like "Thus, it is reasonable to attribute the increase of baseflow and the slowdown of baseflow recession process in autumn and early winter to the enlargement of groundwater storage capacity."

Specific comments: 1. Section 2.3.2. Baseflow separation: It is not clear to me, if the baseflow separation in Fig.4b and Fig.7 are calculated by equations (6)-(7).

2. Section 3.1. Variation of annual streamflow and its components The authors conclude that the changes of air temperature may act as a primary climatic factor for streamflow increase simply based on similar increasing tendency between annual streamflow and annual air temperature, but without any further statistical analysis. I would suggest at least a correlation analysis between precipitation/temperature and streamflow at both annual and seasonal scales.

The statement "..., it is reasonable to attribute annual streamflow increase to the accelerated glacier retreat as the consequence of increasing annual air temperature." (Lines 286-288) is also too arbitrary and lack of more analysis to support. One quick way to check is to see if the lost of the glacier mass can support the increase of the streamfow during the study period.

"It can be further concluded that streamflow is recharged by the increased meltwater from the accelerated glacier retreat which may be partly stored in soil and aquifers in the wide and flat valley (Figure 1b), and subsequently discharge into streams as baseflow. " (Lines 300-303) These are all just deductions or hypothesis without validations.

Section 4. Conclusions

"Moreover, the increase of active groundwater storage in autumn and early winter can partly be attributed to the enlargement of groundwater storage capacity by frozen ground degradation, which can provide storage spaces for increased glacial meltwater." (Lines 404-406). This statement is not supported by solid analysis or materials.

In summary, this manuscript needs more solid quantitative analysis or materials to support the statements on the impacts of glacier or frozen ground degradation on streamflow changes. At least some statistical correlation analyses are needed, e.g., between P/T and streamfow/baseflow, and between P/T and groundwater storage S. Validations for S changes (e.g., well observations or other ways) would be appreciated.

Please also note the supplement to this comment:
https://www.hydrol-earth-syst-sci-discuss.net/hess-2018-541/hess-2018-541-RC1-supplement.pdf

---

## Referee Comment (RC2) · Anonymous Referee #2 · 3 Jan 2019

The authors investigated streamflow changes and active groundwater storage in response to climate warming in a headwater catchment called Yangbajain in the Lhasa River basin on the Tibetan Plateau. The Mann-Kendall test was applied to detect trends of time series. An existing algorithm was adopted to do baseflow separation. The recession flow analysis method was used to determine active groundwater storage. The authors found out that the increase in streamflow is mainly due to glacier meltwater. The increase of annual baseflow is the main cause of the increase in total streamflow.

The manuscript is well written and easy to follow. However, the originality of the study may be weak. The authors used existing methods to analyze data obtained from vari-

ous agencies. In addition, there are some severe problems in the current manuscript. These problems are list below.

Major comments:

(1) This study seems like a case study. All methods used are already existed in the literature and the data were obtained from other agencies. In addition, the method of recession flow analysis may not be appropriate in the study area. As a result, the originality of the study may be weak.

(2) The authors used very simple methods to analyze the complicated system of the Yangbajain catchment. The results are hence questionable.

(3) The meteorological station seems to be a bit too far away from the Yangbajain station. The authors should explain why use the data from the meteorological station are reasonable.

(4) For the equations in the manuscript, if the equation is not derived by the authors, then reference(s) should be added.

(5) In Line 249, the authors stated that "during a period without precipitation and evapotranspiration...". Is this assumption reasonable? A period without precipitation may be reasonable but without evapotranspiration is not. The authors did not add references here or provide an explanation.

(6) The authors stated that the conclusion on streamflow increase (Lines 291-293) is in consistence with Prasch et al. (2013). The manuscript by the authors seems similar to Prasch et al. (2013). Please clarify the differences between the manuscript and the paper by Prasch et al. (2013).

Minor comments:

(1) The authors used "runoff" and "streamflow" simultaneously in the manuscript. Are they means the same thing?

(2) Lines 23-24: Delete this sentence. Such kind of information can be put into the "Study area" section.

(3) Line 38: What is the "mm0.79d-0.21" mean?

(4) Line 216: Is this equation belongs to the MK test or derived by the authors for this study specifically? Why the number on the denominator is 18?

(5) Line 212: This is not a new paragraph.

(6) Line 220: This is not a new paragraph and "When" should be "when".

(7) Lines 234, 244, 255: This is not a new paragraph.

(8) Line 238: The authors stated that the recession flow analysis is widely used, however, only one reference was cited in Line 239. Add more references here.

(9) Line 267: Why a threshold of 0.02 mm/day? Is there any physical background or any derivations of this value?

(10) Lines 291-293: The authors attributed the increase in streamflow to accelerated glacier retreat. Is there any contribution of permafrost degradation in the area due to increased air temperature?

(11) Lines 358-359: Same to Line 38, what are these units mean?

---

## Author Comment (AC1) · 17 Jan 2019

**Response to D. Van Hoy (SC1)**

Comments to the Author

   This paper quantifies streamflow and groundwater changes due to climate change in an alpine region with a large glacier. This type of work is very important and is likely applicable to other mountainous areas (e.g. the Rocky Mountains in North America and the Andes in South America). Overall the methods of this paper are relatively easy to understand. There is a good use of appropriate references throughout the paper including relevant papers at nearby study sites on the Tibetan Plateau. I think the paper is worthy of being published, however there are some issues with grammar and sections where the paper could stand to be reworded to increase readability and be more concise. The paper is also lacking in regards to the site description, explanation of methods, and analysis. For more details, see the comments and questions below.

**Response:** Many thanks for the positive reviews that we received with respect to our paper hess-2018-541 entitled "Quantifying streamflow and active groundwater storage in response to climate warming in an alpine catchment on the Tibetan Plateau". Those comments are all valuable and very helpful for revising and improving our paper. We have addressed the reviewers' concerns and suggestions carefully.

   The major revisions include the clarification of the purpose of the paper, the validity of recession flow analysis, the improvement of the writing, and the thoroughly revision of the Figures. The concept of the active groundwater storage were defined. We described the vegetation (Figure S2) and the rock/soil types throughout the catchment (Figure S5). The coverage of glaciers and frozen ground and the changes of hydro-meteorological data in four sub-catchments of the Lhasa River Basin were introduced in detail to response corresponding questions (Figure S6, S7 and S8; Table S1 and S2). We clarified the type and level of activity of the Damxung-Yangbajain Fault, and what effect this has on groundwater flow (Figure S9 and S10). We quantified the area extent of frozen ground degradation according to the twice map of frozen ground on the Tibetan Plateau. Simultaneously, we are using isotopes to trace the groundwater in the Lhasa River Basin (Figure S4).

   In the following, we provide point-by-point response to each reviewer comment (blue texts are our responses, while black texts are original comments).

   Once again, we appreciate the time you put in reading our manuscript, and the comments were valuable, refreshing, and encouraging. My co-authors and I hope that we have adequately addressed all the review comments.

**Comments/Questions**

1. I would capitalize catchment and station if it has a name before it, i.e. Yangbajain Catchment.

**Response:** Yes, we have corrected it in the revised manuscript, e.g. Yangbajain Catchment/Station, Lhasa River Basin, Damxung-Yangbajain Fault.

2. Does "the wide and flat valley" have a name. As long as there are no other valleys mentioned, I think after you first introduce and describe the valley as wide and flat for the rest of the paper you could just refer to it as the valley.

**Response:** "The wide and flat valley" of the Yangbajain Catchment has not yet been named. There are no other valleys mentioned. We took your suggestion and referred to it as the valley in the rest of the revised manuscript.

3. Throughout the paper sometimes it is referred to as baseflow recession processes and other times it is baseflow recession process. Is there more than one process? What are the process(es) and is the word process even necessary?

**Response:** It is a clerical error for the alternation of baseflow recession processes and baseflow recession process. There is only one process, that is, the baseflow recession. To avoid misunderstanding, we removed the word process(es) from baseflow recession process(es) in the revised manuscript.

4. Since you can separate out baseflow from quickflow using equation 6 and 7, why did you only analyze the base flow recession curves during the fall and early winter and not the entire year. If this is because of the lack of ET or precipitation, then why is there no ET in fall/winter. Is all of the vegetation dead in fall, and what is the vegetation in the area?

**Response:** The annual average hydrograph of the Yangbajain Catchment is like a flood (Figure S1). Generally, the hydrograph shows the rising process of streamflow from January to August, which slightly increases from January to April and rapidly increases from May to August. Thus, we adopt daily streamflow and precipitation records from September to December (the autumn and early winter) over the period 1979-2013 in the catchment, during which the hydrograph with little precipitation usually declines consecutively and smoothly.

[Figure]

Figure S1. The annual average hydrograph of the Yangbajain Catchment from 1979 to 2013.

The catchment has a summer (June-August) monsoon with 73% of the yearly precipitation and only 17% of the yearly precipitation (mainly as snow) occurs in autumn and winter. Since October, the monthly mean air temperature is below 0℃, evaporation is weak and the vegetation is dead especially in winter (Figure S2a). Alpine cold desert soil appears above 5200 m and alpine meadow soil is found on slopes at about 5000 m. It is well vegetated with alpine meadow, alpine steppe, marsh, shrub, etc; meadow and marsh are mainly distributed in the valley and river source (Zhang et al., 2010). The valley in the catchment has a large marsh wetland with an area of 169 km$^2$, which is an excellent pasture in the northern Tibetan Plateau (Figure S2b).

[Figure]

(a) Photographed in winter of 2018

(b)

Figure S2. The (a) vegetation in autumn and winter and (b) marsh wetland in the Yangbajain Catchment.

5. How valid is equation 9 considering that some of the groundwater will go into deeper aquifers, or could potentially come up from deeper aquifers? Are there any areas in the catchment where there is a gradient such that are losing reaches of the stream? Might want to clarify what exactly is meant by active layer storage.

**Response:** In this study, the active groundwater storage $S$ in Equation (9) is deduced from the streamflow $Q$. "The active groundwater storage" is defined as the storage that controls streamflow dynamics assuming that streamflow during rainless periods is a function of catchment storage drainage (see Mobile storage also in Staudinger et al., 2017).

Here, the application of Equation (9) has synthetically included deep groundwater circulation. Because the widespread distribution of hot springs in the Yangbajain Catchment manifest that the groundwater could come up from deeper aquifers and contribute to stream flow (Figure S3).

[Figure]

Figure S3. The hot springs in the Yangbajain Catchment.

In fact, we are using isotopes to trace the groundwater in the Lhasa River Basin especially in the Yangbajain Catchment (see the groundwater sampling points in Figure S4).

Additionally, although there are faults in the catchment, the streams are not cutoff, that is, there are no losing reaches of the stream.

[Figure]

Figure S4. The distribution of water samples points in the Lhasa River Basin.

6. You have a fair description of the valley on the east side of the catchment but no real mention of other prominent features in the catchment. I know this is not a geology paper but if you are going to discuss ground water storage, you should probably give a basic description of the aquifers and rock/soil types throughout the catchment and not just in the valley. I noticed that Figure 3 mentions fractured bedrock but this seems to be the only mention of bedrock in the entire paper. Also it would probably be a good idea to mention what physically separates this catchment (no flow boundaries); i.e., mountains, rivers, etc.; and make sure it is clearly stated in this introduction why you chose this particular catchment.

**Response:** The basic description of the aquifers and rock/soil types throughout the catchment as "The surface of the valley is blanketed by Holocene-aged colluvium, filled with the great thickness of alluvial-pluvial sediments from the south such as gravel, sandy loam, and clay.

Below the colluvium is a large granite batholith of the Eogene. The southeast of the catchment consists of reddish brown and grey green clastic rocks and local volcanic rocks of the Eogene. In the northwest of the catchment, the Nyainqêntanglha Range is mostly composed of biotite plagioclase leptynite, shallow conglomerate and gneiss of the Lower Palaeozoic. Moreover, coarse sandstone and argillaceous siltstone of the Early-Cretaceous are scattered widely in this section (Figure S5) (Jiang et al., 2016)".

We mentioned what physically separates this catchment in the revised manuscript as "The catchment is physically separated by the Nyainqêntanglha Range to the northwest and the Yarlu-Zangbo suture to the south". We also clearly stated why we chose this catchment in the introduction as "The Yangbajain Catchment of the Lhasa River Basin is selected as the study area since it has relatively long-term observation records on the typical ungagged basin of the south-central TP. Moreover, the catchment is experiencing glacier retreat and frozen ground degradation in response to climate warming".

[Figure]

Figure S5. The rock and soil types in the Yangbajain Catchment.

**7.** Make sure that is clear from the abstract not only what you are doing, but how is it different than other studies. Also you might want to include as part of the last sentence what should be a major point in this paper; What impact this has on the population and overall environment of the region? You brought this up in the introduction by talking about water supply but you might consider moving this to the abstract.

**Response:** It was corrected accordingly. We change the last sentence in abstract as "This study provides a perspective to clarify the impact of glacial retreat and frozen ground degradation on hydrological processes, which fundamentally affects the water supply and the mechanisms of

streamflow generation and change".

8. Might want to include some form of outline in your introduction. The introduction is rather long, and it is probably worth another look to make sure that it is as concise and organized as possible.

**Response:** We describes the form of outline in the last paragraph of the introduction as "The paper is structured as follows. The Materials and Methods section includes the study area, data sources and methods. The Results and Discussion section presents the changes in streamflow and its components, climate factors, and glaciers, and analyses the changes in streamflow volume and baseflow recession in response to the changes in active groundwater storage. The conclusions are summarized in the Conclusions section".

9. How much water does the thawing of frozen ground/ permafrost add to the system? Is there a way to quantify this and would it need to be considered in dS/dt?

**Response:** According to the new map of permafrost distribution on the Tibetan Plateau (Zou et al., 2017), the coverages of permafrost and seasonally frozen ground in Lhasa, Pangdo and Tangga sub-catchments are comparable to that in the Yangbajain Catchment; but the coverage of glaciers in the three catchments is far lower than that in the Yangbajain Catchment according to the First Chinese Glacier Inventory (Mi et al., 2002) (Figure S6, Table S1).

The MK test showed that, in all the four catchments, the annual mean air temperature had significant increases at the 1% significance level (Figure S7) while the annual precipitation showed non-significant trends (Table S2). The annual streamflow of the Lhasa, Pangdo and Tangga Catchments all had non-significant trends, while the annual streamflow of Yangbajain Catchment showed an increasing trend at the 5% significance level with a mean rate of about 12.30 mm/10a during the period (Figure S8).

Ye et al. (1999) also stated that when glacier coverage is greater than 5%, glacier contribution to streamflow starts to show up. This indicates that, in the Yangbajain Catchment, the increased streamflow is mainly fed by glacier meltwater rather than frozen ground degradation. It doesn't need to be considered in dS/dt.

[Figure]

Figure S6. The distribution of glaciers and frozen ground in the Lhasa River Basin.

Table S1. The coverage of glaciers and frozen ground in four catchments of the Lhasa River Basin

| Stations | Area (km²) | Coverage (%) | | |
|---|---|---|---|---|
| | | Glaciers | Permafrost | Seasonally frozen ground |
| Lhasa | 26233 | 1.3 | 37 | 63 |
| Pangdo | 16425 | 2.1 | 50 | 50 |
| Tangga | 20152 | 1.7 | 47 | 53 |
| Yangbajain | 2645 | 12.0 | 36 | 64 |

Table S2. Mann-Kendall trend test with trend-free pre-whitening of annual mean air temperature (°C), precipitation (mm) and streamflow (mm) in four sub-catchments of the Lhasa River Basin

| | Air temperature | | Precipitation | | Streamflow | |
|---|---|---|---|---|---|---|
| | $Z_C$ | $\beta$ (°C/a) | $Z_C$ | $\beta$ (mm/a) | $Z_C$ | $\beta$ (mm/a) |
| Lhasa | 6.07** | 0.028 | 1.16 | 1.581 | 1.09 | 1.420 |
| Pangdo | 6.19** | 0.026 | 0.89 | 1.435 | 0.30 | 0.223 |
| Tangga | 7.35** | 0.021 | 1.48 | 2.005 | -0.62 | -0.531 |
| Yangbajain | 4.48** | 0.028 | 1.28 | 2.541 | 2.07* | 1.230 |

[Figure]

Figure S7. Variations of annual mean air temperature in four catchments of the Lhasa River Basin (a: Lhasa; b: Pangdo; c: Tangga; d: Yangbajain).

[Figure]

Figure S8. Variations of annual streamflow in four catchments of the Lhasa River Basin (a: Lhasa; b: Pangdo; c: Tangga; d: Yangbajain).

To quantify the thawing of frozen ground add to the system, air temperatures, soil temperatures at different levels and active layer depths are need to measure. We have been conducting field observations but observation period is still short.

10. It would be a good idea in a future study to include a water budget that includes ET and precipitation.
**Response:** Thanks for your suggestion. In fact, we are doing a work that considers water budget that including ET and precipitation, e.g. the Budyko framework.

11. Is there any snow in the catchment, and does it have any impact on streamflow or is all the meltwater just from the glacier in summer?
**Response:** The precipitation for elevations above about 5,900 m falls in the form of snow and increases the accumulation of glaciers. For elevations below about 5,900 m, the precipitation mainly falls in the form of rainfall and the snow volume is little. There is no runoff in winter due to the dryness of soil. And the spring flood is mostly caused by glacier meltwater rather than snow meltwater, due to less snow volume at lower altitude. So snow has little impact on streamflow.

12. It would be good idea in the future to perform a groundwater tracer study if possible.
**Response:** Thanks for your suggestion. At present, we performs a groundwater tracer study. Figure S4 shows the distribution of groundwater sampling points.

**Detailed Comments**

1. Page 2, line 20 should either be written as an alpine glacier, or alpine glaciers depending on if there are more than one glacier present in the area

**Response:** There are about 319 glaciers in the Yangbajain Catchment. We changed this expression as alpine glaciers.

2. Page 2, lines 28-29 "takes responsibilities to" sounds awkward. I would suggest writing something like climate warming plays a key role in increasing streamflow, or climate warming is partially responsible for increasing streamflow.

**Response:** We changed it as "climate warming plays a key role in increasing streamflow".

3. Page 2, lines 30-31 might read better as, which has led to a loss of over 25% of the total glacier volume.

**Response:** It was corrected accordingly as "which has led to a loss of over 25% of the total glacier volume."

4. Page 2, lines 29-33 it seems that both glacier melt and baseflow are given as the main reason for the increase in streamflow, since the lines use the words "increased streamflow is mainly" and "dominant factor for the increase." It might be best to clarify if the baseflow is from meltwater or if the meltwater only increases streamflow at a certain time of year.

**Response:** It was corrected accordingly. Exactly, the increase in baseflow is the main contributor to streamflow increase. We changes it as "It is concluded that the increase of baseflow is mainly fed by glacier meltwater". The increased meltwater may be partly stored in soil and aquifers in the valley, and subsequently discharge into streams as baseflow.

5. Page 2, line 35 $mm^{0.79}d^{-0.21}/10a$ seem like rather odd units. Where do they come from?

**Response:** Sorry for the error unit in the manuscript. The unit should be [$mm^{0.79}d^{0.21}/10a$]. This is the unit of recession coefficient $K$. We assume nonlinearized outflow from aquifers into streams (i.e. $b{\neq}1$). The fitted slope $b$ is equal to 1.79 through the non-linear least square fit of equation 10 for all data points of $-dy/dt$ versus $y$ in log-log space during the period 1979-2013. The parameters of $K$ and $m$ in equation 8 can be expressed by $a$ and $b$, where $K=1/[a(2-b)]$ and $m=2-b$. The parameter $m$ is equal to 0.21. The units of change in streamflow ($-dy/dt$) and streamflow ($y$) are [$mm/d^2$] and [$mm/d$]. According to the empirical power-law form $-\frac{dy}{dt} = ay^b$, the unit of recession intercept $a$ is [$mm^{-0.79}d^{-0.21}$]. According to $K=1/[a(2-b)]$ ,the unit of recession coefficient $K$ is [$mm^{0.79}d^{0.21}$].

6. Page 3, line 38 might read easier as, which lead to the enlargement of the storage capacity that can accommodate summer rainfall …, which is slowly released into streams in subsequent seasons.

**Response:** It was corrected accordingly as "which lead to the enlargement of the storage capacity that can accommodate summer rainfall and increasing meltwater in the valley, which is slowly released into streams in subsequent seasons."

7. Page 3, line 44 the phrase "way of glacial retreat" would sound better as impact of glacial retreat.

**Response:** It was corrected accordingly as impact of glacial retreat.

8. Page 3, line 51 might read better as, dry periods, giving it a pivotal role…

**Response:** It was corrected accordingly as "dry periods, giving it a pivotal role for water supply for downstream populations, agriculture and industries in these rivers (Viviroli et al., 2007; Pritchard, 2017)".

9. Page 3, lines 51-56 how is the current glacier retreat different than the melt that help sustained the area during dry periods? Were there any particular dry periods that line 51 is refereeing to?

**Response:** Currently, it is still in earlier phase of glacier retreat that the melt remain sustaining the area during dry periods. Generally, the "dry periods" referrers to from November to April.

10. Page 4, line 59 should read, a glacier is known as a "solid reservoir." Not sure if the term solid reservoir actually needs the quotation marks.

**Response:** We change it as "a glacier is known as a "solid reservoir"." According to the usage of the term in published references, we think that the quotation marks are needed.

11. Page 5, line 88 should put the words of the in front of "areal extent".

**Response:** It was corrected accordingly.

12. Page 5, lines 87-91. Sentence order might be better if the portion from the word using to the end of the sentence was moved to just after the word found. It would go something like Evans et al. (2015) found using … on the northern TP that an increase in mean ….

**Response:** It was corrected accordingly. We changes the sentence as "Evans et al. (2015) found using a physically based groundwater model in a headwater catchment of the Heihe River on the northern TP that an increase in mean annual surface temperature of 2°C reduced approximately 28% of the areal extent of permafrost and tripled baseflow contribution to streamflow".

13. Page 5, line 93 capitalize the word basin.

**Response:** It was corrected accordingly as Heihe River Basin.

14. Page 5, line 96 do not need the word the before Artic.

**Response:** It was corrected accordingly as "Arctic rivers".

15. Pages 5-6, line 97-100 might read better if the phrase "based on numerical simulations" was moved to between suggested and that. The sentence would read … suggested based on numerical simulations that ….

**Response:** It was corrected accordingly. We changes the sentence as "Bense et al. (2012) suggested based on numerical simulations that the increasing groundwater storage caused by frozen ground degradation would delay baseflow increase possibly by several decades to centuries".

16. Page 6, line 100 I would change "The slowdown" to A slowdown.
**Response:** It was corrected accordingly as "A slowdown in baseflow recession was found in the northeastern and central TP".

17. Page 6, line 104 remove the ly from qualitatively.
**Response:** It was corrected accordingly as "While, previous qualitative studies were important for understanding the effects of climate warming on hydrological changes in cold alpine catchments".

18. Page 6, line 110 should add the word a before the word warming.
**Response:** It was corrected accordingly as a warming climate.

19. Page 6, line 112 and line 116 insert the before the word catchment.
**Response:** It was corrected accordingly as "at the catchment scale".

20. Page 6, lines 113-119 would switch the order of be and theoretically to read … can theoretically be used …. Also, this is a rather long sentence that could be broken up
**Response:** It was corrected accordingly. We broke up the rather long sentence as "It is difficult to directly measure catchment aquifer storage (Staudinger, 2017; Käser and Hunkeler, 2016) and the GRACE satellites has low spatial resolution in assessing total groundwater storage changes at the catchment scale (Green et al., 2011). An alternative method, namely, recession flow analysis, can theoretically be used to derive the active groundwater storage volume in the phreatic aquifer to reflect frozen ground degradation in a catchment (Brutsaert and Nieber, 1977; Brutsaert et al., 2008)."

21. Page 7, line 121 should either read the non-linearity of the … or a non-linear storage ….
**Response:** We changed it as the non-linearity of the storage discharge…

22. Page 7, line 123 this is not really important for the objectives of this paper but I am curious what the authors would consider to be the complex structure or properties of aquifers in the study area that necessitate a non-linear relationship. Is it simply the presence or areas of frozen ground?
**Response:** For natural conditions,the non-linearity of the storage discharge relationship dominates baseflow recession for most catchments. Linearized outflow from aquifers into streams can be used to simplify the storage discharge relationship in some specific catchments. However, it is complex for underlying surface of the Yangbajain Catchment, including glaciers, permafrost, wetland and fault zone, etc. Assuming nonlinearized outflow is more fitted with the actual storage discharge relationship in the catchment. Here, so we emphasized that the complex structure or properties of aquifers in the catchment should be taken into account.

23. Page 7, line 126 might want to add the word base in front of flow.
**Response:** It was corrected accordingly as "to fit baseflow recession curves".

24. Page 7, line 127-128 I would add an apostrophe 's after Kirchner.
**Response:** It was corrected accordingly as Kirchner's.

25. Page 7, lines 134-135 I would write this as to assess the glacier variations due to climate warming.
**Response:** It was corrected accordingly.

26. Page 7, lines 137-138 should change in climate warming to in a warming climate. Would also change "the water volume changes in the partitioning" to the changes in the partitioning of water volume.
**Response:** It was corrected accordingly.

27. Page 8, line 141 should probably number the last objective to stay consistent.
**Response:** We numbered the last objective as (4) to analyze the impacts of the changes in active groundwater storage on streamflow variation.

28. Page 8, line 146 should change highly to significant.
**Response:** It was corrected accordingly.

29. Page 8, lines 148-152 I would change "the wide and flat valley" to a wide and flat valley. If I understand it correctly the entire valley is in a fault, so my question would be how large is this fault? I would also be interested to know the type and level of activity of the fault, and what effect this has on groundwater flow. I do not think that you need to say that there is flat terrain, since you already called it a wide and flat valley. Also you say thicker aquifers meaning thicker in comparison to what exactly. Finally, I would change the sentence to read the great thickness of Quaternary loose sediment.
**Response:** It was corrected accordingly as the great thickness of Quaternary loose sediment. As the northern boundary of the Yadong-Gulu Rift and the Nyainqentanglha Range, the Damxung-Yangbajain Fault plays a significant role on both rifting of the Damxung-Yangbajain Basin and uplift of the Nyainqentanglha Range (Figure S9 and S10). As a half graben system, the north-south trending Damxung-Yangbajain Fault has a strongly active status to accommodate the east-west extension of the Tibetan Plateau by the ductile normal fault (Cogan et al., 1998), which provides the access for groundwater flow as manifested by the widespread distribution of hot springs (Jiang et al., 2016).

[Figure]

Figure S9. Relief map around Damxung-Yangbajain Fault (see Jiang et al., 2016).

[Figure]

Figure S10. (a) The location, (b) elevation distribution, and (c) glacier and frozen ground distribution in the Yangbajain Catchment.

30. Page 8, lines 152-153 I would change the sentence to read Glaciers cover (or A glacier

covers if it is only one glacier) about 11% of the catchment, making it the most ….

**Response:** We changed the sentence as Glaciers cover about 11% of the catchment, making it the most ….

31. Page 8, lines 154-155 I would move the phrase "according to the First Chinese Glacier Inventory" to the beginning of the sentence and change the rest to read with a majority of glaciers found ….

**Response:** It was corrected accordingly as "According to the First Chinese Glacier Inventory (Mi et al., 2002), the total glacier area was about 316.31 km2 in 1960."

32. Page 8, line 157 I would change ranges to last.

**Response:** It was corrected accordingly as "A majority of glaciers were found along the Nyainqêntanglha Ranges (Figure 1c). Glaciers cover about 11% of the catchment, making it the most glacierized sub-catchment in the Lhasa River Basin."

33. Page 8, line 160 where do the estimates for the area of frozen ground and permafrost come from? Is it from the same map mentioned in the previous line?

**Response:** The estimates for the area of frozen ground and permafrost come from the same map mentioned in the previous line "the new map of permafrost distribution on the TP" (Zou et al., 2017).

34. Page 9, line 163 I would remove the words "The climate in" and just have the sentence start at the second the, and insert the word a in front of semi-arid.

**Response:** It was corrected accordingly as "The catchment is characterized by a semi-arid temperate monsoon climate."

35. Page 9, lines 167-171. I would consider combining the two sentences in these lines to read something like The catchment has a summer (June-August) monsoon with 73% of the yearly precipitation, while the rest of the year is dry with only 1% of the yearly precipitation occurring in winter (December-February).

**Response:** We combined the two sentences. It was corrected accordingly.

36. Page 9, line 172 Where is the number for annual runoff depth coming from. Is it supposed to say runoff depth or streamflow?

**Response:** The number for average annual streamflow comes from daily streamflow data at the Yangbajain Station (4,305 m) during the period 1979-2013. It is supposed to say streamflow. It was corrected accordingly.

37. Page 9, lines 173-178. I would reorganize this paragraph a little to say everything about summer, i.e., where the recharge in summer comes from, and the amount of streamflow first then discuss winter.

**Response:** We reorganized this paragraph as "The average annual streamflow is 277.7 mm, and the intra-annual distribution of streamflow is uneven (Figure 2). In summer, streamflow is recharged mainly by monsoon rainfall and summer meltwater, which accounts for approximately 63% of the yearly streamflow (Figure 2). The streamflow in winter with only 4%

of the yearly streamflow (Figure 2) is only recharged by groundwater, which is greatly affected by the freeze-thaw cycle of frozen ground and the active layer (Liu et al., 2011)."

38. Page 10, line 183 I would change neighbor to adjacent.
**Response:** It was corrected accordingly.

39. Page 10, lines 189-190 I would write the units as ˚C/100m or ˚C per 100m. Also, I would change "with elevation" to for elevations.
**Response:** It was corrected accordingly as "The mean monthly lapse rate is set to 0.44 °C/100m for elevations below 4,965 m and 0.78 °C/100m for elevations above 4,965 m in the catchment (Wang et al., 2015)."

40. Page 10, lines 198-200 I would move the phrase which…trends to between the words test and is at the beginning of the sentence. The sentence would then read (MK) test, which is robust against outliers… is applied to detect ….
**Response:** It was corrected accordingly as "The Mann-Kendall (MK) test, which is robust against outliers and is suitable for data with non-normally distributed or non-linear trends, is applied to detect trends of hydro-meteorological time series (Mann, 1945; Kendall, 1975)."

41. Page 12, line 224 take the s off algorithms.
**Response:** It was corrected accordingly.

42. Page 12, line 225 using the term first filter equation implies that there is another filter equation, but this is the only one given in the paper. Did you mean to say that equation six is the first filter and that equation seven is the second filter equation?
**Response:** There is only one filter equation in the paper. So, we removed the word first from the term first filter equation as "The filter equation is expressed as".

43. Page 12, line 230-231 I would describe α before bt, since it is used in equation six and bt only appears in equation seven. There should also be a semicolon not a period in front of α.
**Response:** It was corrected accordingly as "where $q_t$ and $q_{t-1}$ are the filtered quickflow at time step $t$ and $t$-1, respectively; $Q_t$ and $Q_{t-1}$ are the total runoff at time step $t$ and $t$-1; $\alpha$ is the filter parameter, ranging from 0.9 to 0.95; $b_t$ is the filtered baseflow."

44. Page 12, lines 239-241 I would put the variables in the order that they appear in the equation. Although it is acceptable to start a sentence with the word and I do not think it is necessary for this sentence. I would just start the sentence with the letter K.
**Response:** We described the variables in the order that they appear in the equation. Then we started the sentence with the letter K. The revised sentences are followed as "where $S$ is the volume of active groundwater storage in the catchment aquifers (see in Figure 3). The active groundwater storage $S$ is defined as the storage that controls streamflow dynamics assuming that streamflow during rainless periods is a function of catchment storage drainage (Kirchner, 2009; Staudinger, 2017), abbreviated as groundwater storage in the following context; And $K$ and $m$ are constants depending on the catchment physical characteristics; $K$ is the baseflow

recession coefficient, represented the time scale of the catchment streamflow recession process; *y* is the rate of baseflow in the stream in a catchment."

45. Page 13, lines 256-258 if it takes three days for the signal from a rain storm to peter out why remove only the first two? Also, if you are making use of a filter to separate baseflow then why bother to remove any days?

**Response:** For capturing recession characteristics, baseflow recession events are selected from the streamflow hydrographs that recess markedly for at least 3 days after rainfall ceases. It is an empirical approach for removing storm flow that recession data in the first 2 days are excluded.

46. Pages 13-14, lines 262-265 where does the value for Δycrit come from? I do not think the word meanwhile is needed in this sentence. I am assuming that the process that starts on line 263 is the next step in the overall baseflow process and not something that is done because of Δycrit, so what if anything is done if the value for Δycrit is reached. Also there may be a word missing, such as regression, after the word squares.

**Response:** We deleted the word meanwhile and added the word regression after the word squares. The critical precision threshold Δycrit is used to correct the low value of streamflow data. If the critical precision threshold Δycrit is not added, a large number of low value of streamflow data with observation errors would be contained. Thus, according to plots of -dQ/dt versus Q in log-log space in the Yangbajain Catchment, a threshold Δycrit is selected subjectively to eliminate the impact of low value of streamflow data. The magnitude of this value has little influence on the result of recession flow analysis. If the value for Δycrit is reached, the original data is retained for non-linear least squares regression through the data points. The original data is assigned zero if the value for Δycrit is not reached.

47. Page 14, line 266 I would change "the fixed" to a fixed. How realistic is it to have b fixed as a constant, is there any precedence for having a fixed value for b? What would the effect be if b was not fixed? Is there be any reason you would fix b instead of a?

**Response:** The method of recession analysis proposed by Brutsaert and Nieber (1977) remains one of the few analytical tools for estimating aquifer hydraulic parameters at the field scale and beyond. In the method, the recession hydrograph is examined as *-dQ/dt =f (Q)*, where *Q* is aquifer discharge and *f* is an arbitrary function. The observed function *f* is parameterized through analytical solutions to the one-dimensional Boussinesq equation for unconfined flow in a homogeneous and horizontal aquifer.

What has certainly made this method of analysis alluring is that three well-known analytical solutions to the Boussinesq equation for an unconfined horizontal aquifer (two exact solutions (Boussinesq, 1904; Polubarinova-Kochina, 1962) and one an approximation by linearization (Boussinesq, 1903) can be expressed in the form

$$-dQ/dt = aQ^b \tag{S1}$$

Where *b* is a constant and *a* is a function of the physical dimensions and hydraulic properties of the aquifer (Brutsaert and Nieber, 1977). Plotted as $\log(-dQ/dt)$ versus $\log(Q)$, equation (S1) appears as a straight line with slope *b* and intercept *a*. Theoretically, one can fit a line of slope *b* to recession flow data graphed in this manner and determine aquifer characteristics from the

resulting value of *a*, though care needs to be taken when interpreting plots made from measured data (Rupp and Selker, 2006).

Analytical solutions to various forms of the Boussinesq equation of the recession parameters *a* and *b* in $-dQ/dt=aQ^b$ for a horizontal aquifer are listed in Table S3 (see Rupp and Selker, 2006).

Table S3. Analytical solutions to various forms of the Boussinesq equation for a horizontal aquifer.

| Form of Boussinesq Equation[†] | Time Domain | *b* | $a^‡$ | Parameter Set | Source |
|---|---|---|---|---|---|
| Non-linear | Early | 3 | $\dfrac{1.133}{k\varphi D^3 L^2}$ | (i) | *Polubarinova-Kochina* [1962] |
| Non-linear | Early | 3 | $\dfrac{f_{Lo}}{k\varphi(h_0 - D)^2(h_0 + D)L^2}$ | (ii) | *Lockington* [1997] |
| Non-linear; $k(z)=k_D(z/D)^n$ | Early | 3 | $\dfrac{f_{R1}}{k_D\varphi D^3 L^2}$ | (iii) | *Rupp and Selker* [2005] |
| Non-linear | Late | 3/2 | $\dfrac{4.804k^{1/2}L}{\varphi A^{3/2}}$ | (iv) | *Boussinesq* [1904] |
| Non-linear; $k(z)=k_D(z/D)^n$ | Late | $\dfrac{2n+3}{n+2}$ | $\dfrac{f_{R2}}{\varphi}\left[\dfrac{k_D L^2}{2^n(n+1)D^n A^{n+3}}\right]^{\frac{1}{n+2}}$ | (v) | *Rupp and Selker* [2005] |
| Linearized | Late | 1 | $\dfrac{\pi^2 pkDL^2}{\varphi A^2}$ | (vi) | *Boussinesq* [1903] |

[†]Unless specified, *k* = constant.

\* *k* is the hydraulic conductivity; *p* is a constant approximately 0.3465, which is introduced to compensate for the approximation resulting from the linearization; *D* is the aquifer thickness; *L* is the total length of upstream channels; *φ* is the drainable porosity and *A* is the drainage area.

Theoretically, if *b* was not fixed, we cannot determine the aquifer characteristics from the resulting value of *a* (Rupp and Selker, 2006). Moreover, compared to fixed recession intercept *a*, recession slope *b* is considered to be more stable for recession flow analysis.

48. Section headings for 3.1 and 3.2 the headings only mention streamflow but both sections discuss climate trends. I think you should reconsider the names of these sections. The names could simply just be Annual Variations and Seasonal Variations.
**Response:** We changed the names of sections 3.1 and 3.2 as Annual Variations and Seasonal Variations.

49. Page 14, line 278 add the word a in front of non-significant.
**Response:** It was corrected accordingly as "However, annual precipitation has a non-significant trend during this period (Table 1 and Figure 5b)."

50. Page 14, line 279 I would change the wording to, The similarity in the trends of annual streamflow…. The wording as written is confusing and makes it seem that there was a trend found by plotting temperature vs. streamflow instead of the idea that they both have their own increasing trend with time.
**Response:** It was corrected accordingly as "The similarity in the trends of annual streamflow and annual air temperature indicate that the changes of air temperature may act as a primary

factor for accelerated glacial retreat leading to increasing streamflow."

51. Page 14, lines 282-283. I would rewrite this sentence to read, The significant rise in air temperature has caused a continuous retreat of the glaciers in the catchment. The current wording does not seem to be grammatically correct.
**Response:** It was corrected accordingly.

52. Page 14, line 283 I would remove the word twice and the (I &II volumes) and just write the word Inventory as plural.
**Response:** It was corrected accordingly as "According to the Chinese Glacier Inventories, the total glacial area and volume have decreased by 38.06 $km^2$ (12.0%) and $0.47 \times 10^{10}$ $m^3$ (26.2%) respectively over the past 50 years (Figure 6)."

53. Page 15, line 284 I do not think that you need to give the dates of the glacier inventories again, since they are already in the methods section.
**Response:** It was corrected accordingly.

54. Page 15, line 285 you might want to add the word respectively before the word over to be consistent with the rest of the paper.
**Response:** It was corrected accordingly.

55. Section 3.1, paragraphs 1-2 I would combine these paragraphs. This could be done by combining the last sentence of the first paragraph with the sentence that goes from line 285 to 288. I think that the current first sentence of paragraph two could be eliminated. Starting at the end of line 280 perhaps state something to the effect: may act as a primary factor for accelerated glacial retreat leading to increasing streamflow. Then put the numbers for the decrease in glacier size. Then end the paragraph with the Prasch results. Make sure it is clear what sets your findings in this section apart from the Prasch results.
**Response:** We combined paragraphs 1-2 in Section 3.1 as "The annual streamflow of the Yangbajain Catchment shows an increasing trend at the 5% significance level with a mean rate of about 12.30 mm/10a over the period 1979-2013 (Table 1 and Figure 4a). Meanwhile, annual mean air temperature exhibits an increasing trend at the 1% significance level with a mean rate of about 0.28 °C/10a (Table 1 and Figure 5a). However, annual precipitation has a non-significant trend during this period (Table 1 and Figure 5b). The similarity in the trends of annual streamflow and annual air temperature indicate that the changes of air temperature may act as a primary factor for accelerated glacial retreat leading to increasing streamflow. According to the Chinese Glacier Inventories, the total glacial area and volume have decreased by 38.06 $km^2$ (12.0%) and $0.47 \times 10^{10}$ $m^3$ (26.2%) respectively over the past 50 years (Figure 6). With the nonsignificant increase of annual precipitation, it is reasonable to attribute annual streamflow increase to the accelerated glacier retreat as the consequence of increasing annual air temperature. This conclusion is also consistent with previous results by Prasch et al. (2013), who suggested that glacial meltwater contribution to streamflow would increase in the Yangbajain Catchment together with a significant increase in streamflow and nonsignificant trend in precipitation by quantifying present and future glacier meltwater contribution to runoff."

56. Page 15, line 290 remove the word remain and add the word a in front of significant.

**Response:** It was corrected accordingly.

57. Page 15, line 293-294 put the word the in front of annual mean streamflow. Might want to reword the start of the sentence as According to the baseflow separation method described above …. I would change this to being the second sentence in the paragraph.

**Response:** It was corrected accordingly as "According to the baseflow separation method described above, overall, the annual mean baseflow contributes about 59% of the annual mean streamflow in the catchment."

58. Page 15, line 296 put the word the in front of the word two. I would remove the phrase the result shows that because I think that it is obvious that the rest of sentence comes from the results. Could also change it to the MK test shows that ….

**Response:** It was corrected accordingly as "As annual streamflow increases significantly, it is necessary to analyze to what extent the changes in the two streamflow components lead to streamflow increase. The MK test shows that annual baseflow exhibits a significant increasing trend at the 1% level with a mean rate of about 10.95 mm/10a over the period 1979-2013 (Table 1 and Figure 4b)."

59. Page 15, line 298 change the word this to the.

**Response:** It was corrected accordingly as "The trend is statistically nonsignificant for annual quickflow during the period (Table 1)."

60. Page 15, line 302 add a comma in front of which. I feel like the first part of the sentence before the word which has already been covered by the previous paragraphs.

**Response:** We added a comma in front of which. We deleted repetitive words in the sentence before the word which. The changed sentences are followed as "It can be further concluded that streamflow is recharged by the increased meltwater, which may be partly stored in soil and aquifers in the valley (Figure 1b), and subsequently discharge into streams as baseflow."

61. Page 16, lines 306-307 You say that intra-annual variation is obvious from the hydrograph in Figure 2. The first problem that I have with this is that Figure 2 is not really what I would call a hydrograph, since a hydrograph is typically a line graph of discharge at a location vs. time. Second in a quantitative study such as this I would avoid stating that anything is obvious without applying a statistical test like a coefficient of variation.

**Response:** Thanks for your suggestion. It is really inaccurate to call it a hydrograph. We also removed the word obvious for this sentence. These sentences have been rewritten as "The annual streamflow and baseflow significantly increase due to the rising air temperature over the period 1979-2013. However, there are diverse intra-annual variation characteristics for streamflow and the two streamflow components during the period."

62. Page 16, lines 307-308. I am not sure where you are going with this sentence, and I am not sure if this sentence is even needed. It seems more like the rest of the paragraph is about how streamflow components change based on the season not how they change with magnitude of

streamflow.

**Response:** It's really inappropriate for this sentence to appear here since it does not correspond to the rest of the paragraph. So we deleted this sentence.

63. Page 16, lines 308 this sentence is worded strangely. I have never heard it called a variation trend. What exactly is a streamflow regime?

**Response:** These sentences have been rewritten. Streamflow regime refers to the change of streamflow with time and space.

64. Page 16, line 310 not sure the word however is even necessary.

**Response:** We deleted the word however.

65. Page 16, lines 308-316. Seems like it is getting repetitive and I think this whole section needs to be redone. For example, it says that quick flow shows no trend for all seasons then it says that "streamflow and its two components" have no trend in summer. All seasons would by definition include summer. Another example is saying that baseflow and streamflow have a trend at the 5% level in autumn, winter, and spring; and then going on to state "streamflow increases at the 5% level in autumn" just one sentence later. Maybe try saying all the trends for streamflow in one sentence, then break it into trends for baseflow and quickflow in the next two sentences.

**Response:** The whole section was rewritten as "Streamflow in autumn, winter and spring show increasing trends at least at the 5% significance level (Figure 7c, 7d and 7a), while streamflow in summer has a non-significant trend during this period (Figure 7b). Baseflow increases significantly in autumn, winter and spring (Figure 7c, 7d and 7a). The trend is statistically nonsignificant for baseflow in summer (Figure 7b). Quickflow exhibits nonsignificant trend for all seasons (Table 1)."

66. Page 16, lines 320-321 it has already been stated in the paper that monsoon rainfall is 73% of total precipitation, this does not have to be restated here.

**Response:** We deleted repetitive expression.

67. Page 16, line 322 instead of just saying that the glacier meltwater is considerable is it at all possible to quantify this.

**Response:** Thanks for the nice suggestions. Prasch et al. (2013) quantified rainfall generates about 49% of the streamflow whereas glacier meltwater contributes to max. 11% of the streamflow in the Yangbajain Catchment. The revised sentences are followed as "Compared with monsoon rainfall as the main water source for summer, the corresponding glacial meltwater contribution to the streamflow accounts for max. 11% in the catchment (Prasch et al., 2013)."

68. Page 16, line 323 Is it possible to estimate using a water budget how much of the monsoon rain actually makes it all the way to the stream and how much goes to ET or groundwater?

**Response:** Due to the limited data at present, we cannot estimate using a water budget how much of the monsoon rain actually makes it all the way to the stream or groundwater.

69. Page 16, line 324 remove the word the from the end of the line.

**Response:** It was corrected accordingly as "with thin river valleys".

70. Page 16, line 325 make the word valley plural.

**Response:** It was corrected accordingly.

71. Page 17, line 326 change are to is.

**Response:** It was corrected accordingly as "and then is allowed to rapidly drain through surface layers."

72. Page 17, line 333 change selected to selection.

**Response:** It was corrected accordingly as "Using the data selection procedure mentioned in the section 2.3.3….."

73. Page 17, line 339-340 would change" by all decades of the" to decadely for the, or change the phrase to read by the decade for the 1980's…. I would also perhaps change year-to-year to yearly.

**Response:** It was corrected accordingly. We changed it to the phrase decadely for the and changed year-to-year to yearly. The revised sentences are followed as "With the fixed slope $b$=1.79, the recession coefficient $K$ and groundwater storage $S$ can be quantified decadely for the 1980s, 1990s and 2000s, and yearly from 1979 to 2013. For each decade or year, the recession intercept $a$ could be fitted by the fixed slope $b$=1.79. Then, the values of $K$ and $m$ for each decade or year can be determined with the fitted recession intercept $a$ and the fixed slope $b$. And the groundwater storage $S$ for each decade or year can be directly estimated from the average rate of baseflow during a recession period and the values of $K$ and $m$ through equation (8)."

74. Page 17, line 342 I would change it to read K and m for each year or decade….

**Response:** It was corrected accordingly.

75. Page 17, line 344 Put the word a in front of recession period.

**Response:** It was corrected accordingly.

76. Page 17, line 345-346. If you made change #74 then this last sentence could be eliminated completely, since it does not add anything.

**Response:** We changed #74 and eliminated the last sentence.

77. Page 18, line 348-349 would change "for each decade of the" to just simply the word the.

**Response:** It was corrected accordingly as "Figure 9 shows the non-linear least square fit of equation (10) to the plot of -$dy/dt$ versus y in log-log space for all recession data points of the observation records the 1980s, 1990s and 2000s, respectively."

78. Page 18, line 351 would change it to the word this.

**Response:** It was corrected accordingly.

79. Page 18, lines 356-358. Make sure what you are saying here matches the graph. It seems to me that there were significant increases and decreases in the 1980s, there is a rather large decrease at the beginning of the 1990's not just a slight variation, and it appears that the 2000's have a large increase followed by a decrease before basically leveling off until around 2007 before dropping again.

**Response:** We changed this expression and matched the graph as "The recession coefficient K significantly increases and decreases in the 1980s. There is a rather large decrease at the beginning of the 1990s, and it appears that the 2000's have a large increase followed by a decrease before basically leveling off until around 2007 before dropping again."

80. Page 18, lines 358-360 I would combine these two sentences. To something like but its overall increasing trend of 7.7 $(mm^{0.79}d^{-0.21})$/10a at a significance level of 5% is similar…. I would maybe include somewhere in the paragraph what the decadal trend is numerically rather than just saying it is similar to the yearly trend.

**Response:** It was corrected accordingly. We combined these two sentences as "But its overall increasing trend of 7.70 $(mm^{0.79}d^{0.21})$/10a at a significance level of 5% is similar to the results obtained from decades analysis."

81. Page 18, line 367 change it to this.
**Response:** It was corrected accordingly.

82. Page 19, line 372-375 is not really worded in the most concise manner possible. Might change to Our hypothesis is that increased groundwater storage in autumn and early winter is associated with frozen ground degradation due to rising temperatures during this period.

**Response:** It was corrected accordingly as "Our hypothesis is that increased groundwater storage S in autumn and early winter is associated with frozen ground degradation, which can enlarge groundwater storage capacity (Niu et al., 2016)."

83. Page 19, line 375-376 would change word order to read paths in a glacier fed catchment, which is under-lain by frozen ground.
**Response:** It was corrected accordingly as "Figure 3 depicts the changes of surface flow and groundwater flow paths in a glacier fed catchment, which is under-lain by frozen ground under past climate and warmer climate, respectively."

84. Page 19, line 380-382 perhaps change "to percolate" to the phrase that can percolate. Not sure if the part after the last and in the sentence make sense or is needed. Should not have the word and twice in a list of changes if that is what this is.
**Response:** We changed "to percolate" to the phrase that can percolate and eliminated the part after the last and in the sentence. The revised sentences are followed as "As frozen ground extent continues to decline and active layer thickness continues to increase in the valley, the enlargement of groundwater storage capacity can provide enough storage space to accommodate increasing meltwater, and support more meltwater that can percolate into deeper aquifers rather than surface layers (Figure 3)."

85 Page 19, line 383 change earlier to early.

**Response:** It was corrected accordingly.

86. Page 19, line 385 change increase to increased.

**Response:** It was corrected accordingly.

87. Page 19, last paragraph should include numbers to quantify frozen ground degradation, and if at all possible maybe the thickness of the various layers of aquifers and frozen ground/permafrost.

**Response:** We quantified the area extent of frozen ground degradation which could manifest the corresponding thickness of active layer. The distribution and classification of frozen ground are collected from the map of frozen ground on the Tibetan Plateau (Li and Cheng, 1996) and a new map of permafrost distribution on the Tibetan Plateau (Zou et al., 2017). According to the twice map of frozen ground distribution on the TP (Li and Cheng, 1996; Zou et al., 2017), the areal extent of permafrost in the catchment has decreased by 442 km$^2$ (16%) over the past 22 years; the corresponding areal extent of seasonal frozen ground has increased by 449 km$^2$ (16%) with the degradation of permafrost.

88. Page 20, lines 392-393 change "larger glacierization and large-scale frozen ground" to something like a large glacier and widespread frozen ground coverage.

**Response:** It was corrected accordingly.

89. Page 20 line 397-398 I am assuming by "analyzed the seasonal variations…" you mean the last paragraph of 3.3. If that is the case, I would not really consider this an analysis, but would rather call it a discussion and change the word analyzed to discussed. Figures are referenced but there was really nothing done in the line of statistically quantifying anything like how much frozen ground amount changes seasonally or yearly. Also, it says the "phrase streamflow and its components but the entire last section of the paper only mentions baseflow not flash flow or total streamflow.

**Response:** We changed the word analyzed to the word discussed. And we changed the phrase streamflow and its components to the word baseflow. The changed sentences are followed as "and discussed the seasonal variations of baseflow in response to the changes in active groundwater storage".

90. Page 20, line 402 put a comma in front of which.

**Response:** It was corrected accordingly.

91. Page 20, line 408 states baseflow recession slows down in winter and spring but the study only measured this for fall and the very beginning of winter.

**Response:** We changed the phrase in autumn, winter and spring to the phrase in autumn and early winter.

92. Page 21, line 410 add the word a in front of the word warming.

**Response:** It was corrected accordingly.

**Figure based comments**

1. Figure 1 make sure that you use the highest resolution figure possible, so that the legends remain readable. Since the figure is on a page by itself you might want to reduce white space by making the figure take up as much room as possible.

**Response:** It was corrected accordingly.

[Figure]

**Figure 1.** (a) The location, (b) elevation distribution, and (c) glacier and frozen ground distribution (Zou et al., 2017) in the Yangbajain Catchment of the Lhasa River Basin in the TP.

2. Figures 2, 4, and 7 is this really runoff depth or is it streamflow. On page 7, line 75 it mentions streamflow not runoff depth when it references figure 2. Same thing on page 14, lines 274-276 for figure 4a; and on page 16, lines 309-310 and 316 for figure 7. To me runoff depth implies overland flow, which can be highly variable over a catchment and is hard to measure as a depth.

**Response:** All "runoff depth" have been replaced by "streamflow" in lines and figures.

[Figure]

Figure 2. Seasonal variation of streamflow (R), mean air temperature (T), and precipitation (P) in the Yangbajain Catchment.

[Figure]

Figure 4. Variations of annual (a) streamflow and (b) baseflow from 1979 to 2013.

[Figure]

Figure 7. Variations of seasonal streamflow and baseflow in (a) spring, (b) summer, (c) autumn, and (d) winter from 1979 to 2013.

3. Figure 2 I would split this into a 2a and 2b with 2b just having temperature. Alternatively, the two axis on the right hand side could be combined, since the numbers are almost the same for both.

**Response:** We combined the two axis on the right hand side (see the response of figure based comments 2).

4. Figure 3, I would change the color of either the surface flow paths or the ground water flow paths because it is hard to tell that there a darker blue when groundwater covers them up. Not sure if surface water flow path arrow is even necessary.

**Response:** We changed the color of the groundwater flow paths as red. It is necessary for

surface water flow path arrow to compare the quantity of surface runoff and groundwater runoff.

[Figure]

**Figure 3.** Diagram depicting surface flow and groundwater flow due to glacier melt and frozen ground thaw under (a) past climate and (b) warmer climate. Blue lines with arrows are conceptual surface flow paths. Red lines with arrows are conceptual groundwater flow paths (after Evans and Ge. (2017)).

5. Figure 7 might want to change one of the lines in figure to a different color, since color is allowed in this journal. The lines in figure 7d in particular are close together and it is hard to differentiate the two since they are both black.

**Response:** We changed the color of baseflow lines as blue (see the response of figure based comments 2).

**References:**

Biswal, B., & Marani, M.: Geomorphological origin of recession curves, Geophysical Research Letters, 37, L24403, 2010.

Boussinesq, J.: Sur le de´bit, en temps de se´cheresse, d'une source alimente´e par une nappe d'eaux d'infiltration, C. R. Hebd, Seanc. Acad. Sci. Paris, 136, 1511-1517, 1903.

Boussinesq, J.: Recherches the´oriques sur l'e´coulement des nappes d'eau infiltre´es dans le sol et sur de´bit de sources, J. Math. Pures Appl., 10, 5-78, 1904.

Brutsaert, W.: Hydrology: An Introduction. Cambridge University Press, New York, USA, 2005.

Brutsaert, W., and Nieber, J. L.: Regionalized drought flow hydrographs from a mature glaciated plateau, Water Resources Research, 13(3), 637-643, 1977.

Buttle, J. M.: Mediating stream baseflow response to climate change: the role of basin storage, Hydrological Processes, 32(1), doi:10.1002/hyp.11418, 2017.

Cogan, M. J., Nelson, K. D., and Kidd, W. S. F., et al.: Shallow structure of the Yadong-Gulu rift, southern Tibet, from refraction analysis of project indepth common midpoint data, Tectonics, 1998, 17.

Jiang, W., Han, Z., Zhang, J., and Jiao, Q.: Stream profile analysis, tectonic geomorphology and neotectonic activity of the Damxung-Yangbajain Rift in the south Tibetan Plateau, Earth Surface Processes & Landforms, 41(10), 1312-1326, doi:10.1002/esp.3899, 2016.

Li, S. and Cheng, G.: Map of Frozen Ground on Qinghai-Xizang Plateau, Gansu Culture Press,

Lanzhou, 1996.

Kirchner, J.W.: Catchments as simple dynamical systems: catchment characterization, rainfall-runoff modeling, and doing hydrology backward, Water Resources Research, 45, W02429, doi:10.1029/2008WR006912, 2009.

Mi, D. S., Xie, Z. C., and Luo, X. R.: Glacier Inventory of China (volume XI: Ganga River drainage basin and volume XII: Indus River drainage basin). Xi'an Cartographic Publishing House, Xi'an, pp. 292-317 (In Chinese), 2002.

Prasch, M., Mauser, W., and Weber, M.: Quantifying present and future glacier melt-water contribution to runoff in a central Himalayan river basin, Cryosphere, 7(3), 889-904, doi:10.5194/tc-7-889-2013, 2013.

Polubarinova-Kochina, P. Y.: Theory of Ground Water Movement, 613 pp., Princeton Univ. Press, Princeton, N. J, 1962.

Rupp, D. E., and J. S. Selker.: Information, artifacts, and noise in dQ/dt-Q recession analysis, Adv. Water Resour., 29, 154-160, 2006.

Staudinger, M., Stoelzle, M., Seeger, S., Seibert, J., Weiler, M., and Stahl, K.: Catchment water storage variation with elevation, Hydrological Processes, 31(11), doi:10.1002/hyp.11158, 2017.

Ye, B. S., Yang, D. Q., Zhang, Z. L., and Kane, D. L.: Variation of hydrological regime with permafrost coverage over Lena basin in Siberia, Journal of Geophysical Research Atmospheres, 114, ZD07102, doi: 10.1029/2008JD010537, 2009.

Zhang, Y., Wang, C., and Bai, W., et al.: Alpine wetland in Lhasa River Basin, China, J Geogr Sci, 20(3): 375-388, 2010.

Zou, D., Zhao, L., Sheng, Y., and Chen, J., et al.: A new map of permafrost distribution on the Tibetan Plateau, The Cryosphere, 11, 2527-2542, doi: 10.5194/tc-11-2527-2017, 2017.

---

## Author Comment (AC2) · 24 Jan 2019

**Response to the anonymous reviewer 2# (RC2)**

The authors investigated streamflow changes and active groundwater storage in response to climate warming in a headwater catchment called Yangbajain in the Lhasa River basin on the Tibetan Plateau. The Mann-Kendall test was applied to detect trends of time series. An existing algorithm was adopted to do baseflow separation. The recession flow analysis method was used to determine active groundwater storage. The authors found out that the increase in streamflow is mainly due to glacier meltwater. The increase of annual baseflow is the main cause of the increase in total streamflow. The manuscript is well written and easy to follow. However, the originality of the study may be weak. The authors used existing methods to analyze data obtained from various agencies. In addition, there are some severe problems in the current manuscript. These problems are list below.

**Response:** Good comments and they are valuable and very helpful for revising and improving our paper. In fact, it has always been an important issue to find a linkage between the natural process complexity and the process simplicity for model conceptualization (Sivapalan, 2003). As indicated by Sivapalan (2003), it is widely accepted to infer model structures and conceptualization using available data (e.g., rainfall and runoff).

Here in this study, we use relatively simple quantitative models due to the limited data in the hydrological observations of the Tibetan Plateau. For example, the recession flow analysis and theory is proposed by Brutsaert and Nieber (1977) with physical considerations based on hydraulic groundwater theory. It can theoretically reflect the changes in subsurface storage for indirect detection of permafrost change at the catchment scale (Brutsaert, 2005; Brutsaert et al., 2008; Lyon et al., 2009). The main strength of the recession flow analysis is its ability to capture the integral effect of the process complexity present in a catchment's subsurface based on the catchment's streamflow recession characteristics with a much simpler set of equations than if all the complexity were modelled in detail (Lyon and Destouni, 2010). Future work is needed to test this method in other catchments containing permafrost to determine its general applicability across different geomorphologic and climatic settings. Moreover, it is difficult to establish hydraulic groundwater model in the Yangbajain Catchment with the limited data of hydrological observations and the complex underlying critical zone structure (e.g., glaciers, permafrost and fault zone). Thus, we adopt the recession flow analysis as a lumped model for describing the characteristics of streamflow recession and the properties of aquifers in a catchment as a whole.

The major revisions include the suitability of recession flow analysis, the feasibility of several simple combined methods in the complex catchment with limited available data, the improvement of the writing, and the thoroughly revision of the paper. We explained the reasonability of using the data from the meteorological station at Damxung. We clarified the differences between our paper and the paper by Prasch et al. (2013). We compared the coverages of frozen ground and glaciers in four sub-catchments of the Lhasa River Basin to confirm that, in the Yangbajain Catchment, the increased streamflow is mainly fed by glacier meltwater rather than frozen ground degradation (Table S1 and S2).

In the following, we provide point-by-point response to each reviewer comment (blue texts are our responses, while black texts are original comments).

Once again, we appreciate the time you put in reading our manuscript, and the comments

were valuable. My co-authors and I hope that we have adequately addressed all the review comments.

**Major comments:**

1. This study seems like a case study. All methods used are already existed in the literature and the data were obtained from other agencies. In addition, the method of recession flow analysis may not be appropriate in the study area. As a result, the originality of the study may be weak.

**Response:** One of the limiting factors in the hydrological study of the TP is the availability of observations (Cuo et al., 2014). Due to the harsh natural environmental conditions, many areas on the TP are not accessible, and in situ field measurement stations are difficult to establish and maintain (Cuo et al., 2015). In the Yangbajain Catchment, the available data are relatively less and the study is relatively weak. We have been conducting field observations, such as air temperatures, soil temperatures at different altitudes and active layer depths, etc. But observation period is too short to support long-term data analysis. So we collect the existing data (climatic factors, discharge, glacier, etc) from national database and other agencies as much as possible and adopt suitable methods to reveal the mechanisms of runoff changes based on the available data.

It has always been an important issue to find a linkage between the natural process complexity and the process simplicity for model conceptualization (Sivapalan, 2003). As indicated by Sivapalan (2003), it is widely accepted to infer model structures and conceptualization using available data (e.g., rainfall and runoff). At local scales, such as mountain slopes or experimental small catchments, "surgical approach" is often adopted to acquire the storage and streamflow data. At larger scales, however, direct observations of permafrost change especially on the TP are difficult to perform and there is a need for indirect detection methods of permafrost change and its effects on runoff response at larger scales (Lyon et al., 2009). The recession flow analysis, can theoretically be used to derive the active groundwater storage volume to reflect frozen ground degradation at the catchment scale (Brutsaert and Nieber, 1977; Brutsaert, 2005; Brutsaert et al., 2008). In addition, the recession flow analysis is more like a lumped method for describing the characteristics of streamflow recession and the properties of aquifers in a catchment as a whole, which is more suitable for larger scales rather than local scales.

The recession flow analysis and theory is proposed by Brutsaert and Nieber (1977) with physical considerations based on hydraulic groundwater theory. Recession flow analysis for forecasting drought flows and investigating the groundwater flow regime in basins has over a century-long history (Hall, 1968; Tallaksen, 1995). It remains one of the few analytical tools for estimating aquifer hydraulic parameters at the field scale and beyond (Rupp and Selker, 2006). The main strength of the recession flow analysis is its ability to capture the integral effect of the process complexity present in a catchment's subsurface on the catchment's streamflow recession characteristics with a much simpler set of equations than if all the complexity were modelled in detail (Lyon and Destouni, 2010). For example, permafrost thawing is a complex and variable process. As such, a few local observations of permafrost and permafrost thawing may not be able to capture accurately the overall catchment-scale changes (Lyon and Destouni, 2010). Recession flow analysis provides an elegant methodology for reflecting catchment-scale permafrost changes over time (Lyon et al., 2009).

Many previous studies show that the recession flow analysis can be appropriated at the catchment scale with widespread permafrost (Lyon et al., 2009; Lyon and Destouni, 2010; Sjöberg et al., 2013; Lin and Yeh., 2017). Lyon et al. (2009) used recession flow analysis based on a long-term streamflow record to estimate permafrost thawing at an average rate of about 0.9 cm/yr during the past 90 years in the sub-arctic Abiskojokken Catchment of northern Sweden. Lyon and Destouni (2010) tested the ability of recession flow analysis to reflect thawing of permafrost at the catchment scale for the well-studied Yukon River Basin covering large portions of Alaska, USA and parts of Canada, which the changes in the recession flow properties detected in the Yukon River Basin agree well with observations of permafrost thawing across central Alaska. Sjöberg et al. (2013) illuminated the potential for recession flow analysis based on hydrologic observations to monitor changes in catchment scale permafrost. Further, it opens the door for research to isolate the mechanisms behind the different trends observed and to gauge their ability to reflect actual permafrost conditions at the catchment scale. Lin and Yeh. (2017) adopted recession flow analysis by assuming linearized outflow from aquifers into streams to estimate groundwater storage in northern Taiwan. On the basis of Lyon et al. (2009) and Lin and Yeh. (2017), we further develop this approach by assuming nonlinear outflow and find the increase of groundwater storage in autumn and early winter. This conclusion has been validated by the GRACE satellites to assess total groundwater storage changes at the catchment scale (See response the specific comments 3 to the anonymous reviewer 1#).

Even so, we certainly hope that we get more data to do in-depth analysis at local scale in the future.

2. The authors used very simple methods to analyze the complicated system of the Yangbajain catchment. The results are hence questionable.
**Response:** A very good comment. Under natural conditions, the hydrological process complexity underlying catchment responses and the complex structures and properties of catchment aquifers are often characterized by strong spatial-temporal distribution and vertical zoning characteristics especially in cold alpine catchments (Guan et al., 1984). In fact, it has always been an important issue to find a linkage between the natural process complexity and the process simplicity for model conceptualization (Sivapalan, 2003). As indicated by Sivapalan (2003), it is widely accepted to infer model structures and conceptualization using available data (e.g., rainfall and runoff).
In fact, it is limited for field hydrological observations in most cold-alpine catchments. For example, there are only 40 hydrological stations (discharge, Precipitation) in the whole Tibet Autonomous Region. That's how it requires us to use the limited available data and select feasible methods for revealing the physical mechanism of catchment. The method of recession flow analysis can be able to estimate groundwater storage and capture accurately the overall catchment-scale permafrost changes only based on a long-term streamflow record (Lyon et al., 2009; Lyon and Destouni, 2010; Sjöberg et al., 2013).

3. The meteorological station seems to be a bit too far away from the Yangbajain station. The authors should explain why use the data from the meteorological station are reasonable.
**Response:** There are only three national meteorological stations in the Lhasa River Basin

(Damxung Station, Lhasa Station and Medro Gongkar Station). The monthly meteorological data at the Damxung station (4,289 m), which is about 69.2 km to Yangbajain station (Figure S1). We adopted the method of meteorological data extrapolation based on the limited meteorological data. Moreover, the method of meteorological data extrapolation by assuming a linear lapse rate has been successfully used in the Yangbajain Catchment by Prasch et al. (2013).

The specific steps of meteorological data extrapolation are as follows: according to the surface elevation $z_s$ (m) of the Damxung station and every grid cell surface elevation $z_{gc}$ (m) of the Yangbajain catchment with cell size of 1km×1km, the every grid cell air temperature $T_{gc}$ (°C) in the catchment is determined by extrapolating the air temperature $T_s$ (°C) of the Damxung station, assuming a linear lapse rate $r$ (°C/100 m) by Equation. (S1)

$$T_{gc} = T_s - \gamma(z_{gc} - z_s) \qquad (S1)$$

In this study, the mean monthly lapse rate is set to 0.44 °C/100 m with elevation below 4,965 m and 0.78 °C/100 m with elevation above 4,965 m in the catchment (Wang et al., 2015). The air temperature $T$ (°C) over the entire area is the mean grid cell air temperature $T_{gc}$ (°C) in the catchment.

4. For the equations in the manuscript, if the equation is not derived by the authors, then reference(s) should be added.

**Response:** Thanks for your nice suggestions. We added the reference(s) for the equations. For examples, the revised sentences are follows as

"Mann (1945) and Kendall (1975) have documented that when n ≥ 8, the mean of s is zero, the variance of s is proposed by the equation (3)",

"Substitution of equation (8) in equation (9) yields (Brutsaert and Nieber, 1977)" and "The parameters of K and m in equation (8) can be expressed by a and b, where $K=1/[a(2-b)]$ and $m=2-b$ (Gao et al., 2017)."

5. In Line 249, the authors stated that "during a period without precipitation and evapotranspiration...". Is this assumption reasonable? A period without precipitation may be reasonable but without evapotranspiration is not. The authors did not add references here or provide an explanation.

**Response:** Sorry for the misleading expression in the manuscript. Actually, we mean that evapotranspiration is weak and can be neglected during autumn and winter. And we added the references. In our paper, "the active groundwater storage" is defined as the deep storage that controls streamflow dynamics assuming that streamflow during rainless periods is a function of catchment storage drainage (see Mobile storage also in Staudinger et al., 2017). Since October, the monthly mean air temperature is below 0°C (Figure S1), evaporation is weak and the vegetation is dead especially in winter (Figure S2a). The reference evapotranspiration is simply assumed to be 0 mm when the monthly air temperature is less than or equal to 0°C (Gao et al., 2007). During autumn and winter, seasonally frozen soil in the catchment (Figure S2b) servers as an impermeable barrier to the deep evaporation, under which evaporation can be ignored.

The revised sentences are followed as "During a period when precipitation and evaporation can be ignored, the flow in a stream can be assumed to depend solely on the groundwater

storage from the upstream aquifers (Brutsaert, 2008; Lin and Yeh, 2017)."

[Figure]

Figure S1. Seasonal variation of streamflow (*R*), mean air temperature (*T*), and precipitation (*P*) in the Yangbajain Catchment.

(a) vegeration

(b) soil

[Figure]

Figure S2. The landscape in autumn and winter in the Yangbajain Catchment (Photographed in winter of 2018).

6. The authors stated that the conclusion on streamflow increase (Lines 291-293) is in consistence with Prasch et al. (2013). The manuscript by the authors seems similar to Prasch et al. (2013). Please clarify the differences between the manuscript and the paper by Prasch et al. (2013).

**Response:** Prasch et al. (2013) present a model-based analysis to assess the changes of river runoff and ice-melt contribution to river runoff under past and future climatic conditions for the Lhasa River Basin, which focus on how the river runoff changes in the past and future.

We use available data based on several combined methods not only to analysis the changes of streamflow and the two streamflow components (base flow and quick flow) and they are only the results of climate changes. But also we hope to explore how the streamflow as well as the base flow have been changed under the warming climate.

The interesting findings include that: (1) we found the increase of streamflow is attributed to the accelerated glacier retreat due to increased air temperature; (2) Moreover, through recession flow analysis, the increase of active groundwater storage in autumn and early winter can explain why baseflow volume increases.

**Minor comments:**

1. The authors used "runoff" and "streamflow" simultaneously in the manuscript. Are they means the same thing?

**Response:** The "runoff" and "streamflow" means the same thing. It is more suitable to say streamflow in the paper. All "runoff" have been replaced by "streamflow" in lines and figures.

2. Lines 23-24: Delete this sentence. Such kind of information can be put into the "Study area" section.

**Response:** We deleted the sentence "The catchment is characterized by… ".

3. Line 38: What is the "mm0.79d-0.21" mean?

**Response:** Sorry for the error unit in the manuscript. It should be [$mm^{0.79}d^{0.21}$]. This is the unit of recession coefficient $K$. We assume nonlinearized outflow from aquifers into streams (i.e. $b \neq 1$). The fitted slope $b$ is equal to 1.79 through the non-linear least square fit of equation 10 for all data points of $-dy/dt$ versus $y$ in log-log space during the period 1979-2013. The parameters of $K$ and $m$ in equation 8 can be expressed by $a$ and $b$, where $K=1/[a(2-b)]$ and $m=2-b$. The parameter $m$ is equal to 0.21. The units of change in streamflow ($-dy/dt$) and streamflow ($y$) are [$mm/d^2$] and [$mm/d$]. According to the empirical power-law form $-\frac{dy}{dt} = ay^b$, the unit of recession intercept $a$ is [$mm^{-0.79}d^{-0.21}$]. According to $K=1/[a(2-b)]$ ,the unit of recession coefficient $K$ is [$mm^{0.79}d^{0.21}$].

4. Line 216: Is this equation belongs to the MK test or derived by the authors for this study specifically? Why the number on the denominator is 18?

**Response:** This equation belongs to the MK test. Mann (1945) and Kendall (1975) have documented that when $n \geq 8$, the statistic $S$ is approximately normally distributed with the mean and the variance as follows:

$$E(S) = 0$$

$$Var(S) = \frac{n(n-1)(2n+5) - \sum_t t(t-1)(2t+5)}{18}$$

where $n$ is the sequence length, $t$ is the extent of any given tie and represents the sum over all ties.

Detailed details of the process can be found in the references (Mann, 1945; Kendall, 1975).

5. Line 212: This is not a new paragraph.
**Response:** It was corrected accordingly. We deleted the blank space before the paragraph.

6. Line 220: This is not a new paragraph and "When" should be "when".
**Response:** It was corrected accordingly. We deleted the blank space before the paragraph.

7. Lines 234, 244, 255: This is not a new paragraph.
**Response:** It was corrected accordingly. We deleted the blank space before the paragraph.

8. Line 238: The authors stated that the recession flow analysis is widely used, however, only one reference was cited in Line 239. Add more references here.
**Response:** It was corrected accordingly. We added more references in this sentence as "The method of recession flow analysis is widely used to investigate the baseflow recession characteristics and the storage discharge relationship of catchments (Lyon et al., 2009; Lyon and Destouni, 2010; Sjöberg et al., 2013; Lin and Yeh., 2017; Gao et al., 2017)."

9. Line 267: Why a threshold of 0.02 mm/day? Is there any physical background or any derivations of this value?
**Response:** The critical precision threshold Δycrit of 0.02 mm/day is used to correct the low value of streamflow data. If the critical precision threshold Δycrit is not added, a large number of low value of streamflow data with observation errors would be contained. Thus, according to plots of -dQ/dt versus Q in log-log space in the Yangbajain Catchment, a threshold Δycrit is selected subjectively to eliminate the impact of low value of streamflow data. The magnitude of this value has little influence on the result of recession flow analysis.

10. Lines 291-293: The authors attributed the increase in streamflow to accelerated glacier retreat. Is there any contribution of permafrost degradation in the area due to increased air temperature?
**Response:** According to the new map of permafrost distribution on the Tibetan Plateau (Zou et al., 2017), the coverages of permafrost and seasonally frozen ground in Lhasa, Pangdo and Tangga sub-catchments are comparable to that in the Yangbajain Catchment; but the coverage of glaciers in the three catchments is far lower than that in the Yangbajain Catchment according to the First Chinese Glacier Inventory (Mi et al., 2002) (Figure S3, Table S1).

The MK test showed that, in all the four catchments, the annual mean air temperature had significant increases at the 1% significance level (Figure S4) while the annual precipitation showed non-significant trends (Table S2). The annual streamflow of the Lhasa, Pangdo and Tangga Catchments all had non-significant trends, while the annual streamflow of Yangbajain Catchment showed an increasing trend at the 5% significance level with a mean rate of about 12.30 mm/10a during the period (Figure S5).

Ye et al. (1999) also stated that when glacier coverage is greater than 5%, glacier contribution to streamflow starts to show up. This indicates that, in the Yangbajain Catchment, the increased streamflow is mainly fed by glacier meltwater rather than frozen ground degradation.

[Figure]

Figure S3. The distribution of glaciers and frozen ground in the Lhasa River Basin.

Table S1. The coverage of glaciers and frozen ground in four sub-catchments
of the Lhasa River Basin

| Stations | Area (km²) | Coverage (%) | | |
|---|---|---|---|---|
| | | Glaciers | Permafrost | Seasonally frozen ground |
| Lhasa | 26233 | 1.3 | 37 | 63 |
| Pangdo | 16425 | 2.1 | 50 | 50 |
| Tangga | 20152 | 1.7 | 47 | 53 |
| Yangbajain | 2645 | 12.0 | 36 | 64 |

Table S2. Mann-Kendall trend test with trend-free pre-whitening of annual mean air
temperature (°C), precipitation (mm) and streamflow (mm) in four catchments of the Lhasa
River Basin

| | Air temperature | | Precipitation | | Streamflow | |
|---|---|---|---|---|---|---|
| | $Z_C$ | $\beta$ (°C/a) | $Z_C$ | $\beta$ (mm/a) | $Z_C$ | $\beta$ (mm/a) |
| Lhasa | 6.07** | 0.028 | 1.16 | 1.581 | 1.09 | 1.420 |
| Pangdo | 6.19** | 0.026 | 0.89 | 1.435 | 0.30 | 0.223 |
| Tangga | 7.35** | 0.021 | 1.48 | 2.005 | -0.62 | -0.531 |
| Yangbajain | 4.48** | 0.028 | 1.28 | 2.541 | 2.07* | 1.230 |

[Figure]

Figure S4. Variations of annual mean air temperature in four catchments of the Lhasa River Basin (a: Lhasa; b: Pangdo; c: Tangga; d: Yangbajain).

[Figure]

Figure S5. Variations of annual streamflow in four catchments of the Lhasa River Basin (a: Lhasa; b: Pangdo; c: Tangga; d: Yangbajain).

11. Lines 358-359: Same to Line 38, what are these units mean?
**Response:** See the response for Minor comments 3.

**References:**

Brutsaert, W., and Nieber, J. L.: Regionalized drought flow hydrographs from a mature glaciated plateau, Water Resources Research, 13(3), 637-643, 1977.

Brutsaert, W.: Hydrology: An Introduction. Cambridge University Press, New York, USA, 2005.

Brutsaert, W.: Long-term groundwater storage trends estimated from streamflow records: Climatic perspective, Water Resources Research, 44(2), 114-125, doi:10.1029/2007WR006518, 2008.

Cuo, L., Zhang, Y. X., Zhu, F. X., and Liang, L. Q.: Characteristics and changes of streamflow on the Tibetan Plateau: A review, Journal of Hydrology Regional Studies, 2, 49-68, doi:10.1016/j.ejrh.2014.08.004, 2014.

Cuo, L., Zhang, Y., Bohn, T.J., et al.: Frozen soil degradation and its effects on surface hydrology in the northern Tibetan Plateau, Journal of Geophysical Research: Atmospheres, 120(16), 8276-8298, 2015.

Gao, M., Chen, X., Liu, J., Zhang, Z., and Cheng, Q.: Using two parallel linear reservoirs to express multiple relations of power-law recession curves, Journal of Hydrologic Engineering, 04017013, doi:10.1061/(ASCE)HE.1943-5584.0001518, 2017.

Gao, G., D. Chen, C. Y. Xu, and E. Simelton.: Trend of estimated actual evapotranspiration over China during 1960–2002, J. Geophys. Res., 112, D11120, doi:10.1029/2006JD008010, 2007.

Guan, Z., Chen, C., and Ou, Y.: Rivers and Lakes in Tibet, Beijing, Science Press, pp238, 1984 (In Chinese).

Hall, F. R.: Base-flow recessions: A review, Water Resour. Res., 4, 973-983, 1968.

Kendall, M. G.: Rank Correlation Methods, 4th ed, Charles Griffin, London, pp. 196, 1975.

Lin, K. T., and Yeh, H. F.: Baseflow recession characterization and groundwater storage trends in northern Taiwan, Hydrology Research, 48(6), 1745-1756, 2017.

Li, S. and Cheng, G.: Map of Frozen Ground on Qinghai-Xizang Plateau, Gansu Culture Press, Lanzhou, 1996.

Lyon, S. W., and Destouni, G.: Changes in catchment-scale recession flow properties in response to permafrost thawing in the Yukon River basin, International Journal of Climatology, 30(14), 2138-2145, doi:10.1002/joc.1993, 2010.

Lyon, S. W., Destouni, G., Giesler, R., Humborg, C., Mörth, M., and Seibert, J., et al.: Estimation of permafrost thawing rates in a sub-arctic catchment using recession flow analysis, Hydrology and Earth System Sciences Discussions, 13(5), 595-604, 2009.

Mann, H.: Non-parametric test against trend, Econometrica, 13, 245-259, 1945.

Mi, D. S., Xie, Z. C., and Luo, X. R.: Glacier Inventory of China (volume XI: Ganga River drainage basin and volume XII: Indus River drainage basin). Xi'an Cartographic Publishing House, Xi'an, pp. 292-317 (In Chinese), 2002.

Prasch, M., Mauser, W., and Weber, M.: Quantifying present and future glacier melt-water contribution to runoff in a central Himalayan river basin, Cryosphere, 7(3), 889-904, doi:10.5194/tc-7-889-2013, 2013.

Rupp, D. E., Selker, J. S.: On the use of the Boussinesq equation for interpreting recession

hydrographs from sloping aquifers, Water Resources Research, 42(12):1675-1679, 2006.

Sivapalan, M.: Process complexity at hillslope scale, process simplicity at the watershed scale: is there a connection? Hydrological Processes, 17(5), 2010.

Sjöberg, Y., Frampton, A., and Lyon, S. W.: Using streamflow characteristics to explore permafrost thawing in northern Swedish catchments, Hydrogeology Journal, 21, 121-131, 2013.

Staudinger, M., Stoelzle, M., Seeger, S., Seibert, J., Weiler, M., and Stahl, K.: Catchment water storage variation with elevation, Hydrological Processes, 31(11), doi:10.1002/hyp.11158, 2017.

Tallaksen, L. M.: A review of baseflow recession analysis, J. Hydrol., 165, 349-370, 1995.

Wang, S., Liu, S. X., Mo, X. G., Peng, B., Qiu, J. X., Li, M. X., Liu, C. M., Wang, Z. G., and Bauer-Gottwein, P.: Evaluation of remotely sensed precipitation and its performance for streamflow simulations in basins of the southeast Tibetan Plateau, Journal of Hydrometeorology, 16(6), 342-354, doi:10.1175/JHM-D-14-0166.1, 2015.

Ye, B. S., Yang, D. Q., Zhang, Z. L., and Kane, D. L.: Variation of hydrological regime with permafrost coverage over Lena basin in Siberia, Journal of Geophysical Research Atmospheres, 114, ZD07102, doi: 10.1029/2008JD010537, 2009.

Zou, D., Zhao, L., Sheng, Y., and Chen, J., et al.: A new map of permafrost distribution on the Tibetan Plateau, The Cryosphere, 11, 2527-2542, doi:10.5194/tc-11-2527-2017, 2017.

---

## Author Comment (AC3) · 25 Jan 2019

Please refer to the attached pdf

Please also note the supplement to this comment:
https://www.hydrol-earth-syst-sci-discuss.net/hess-2018-541/hess-2018-541-AC3-supplement.pdf